# The cellular architecture of memory modules in *Drosophila* supports stochastic input integration

Omar A Hafez[1†‡], Benjamin Escribano[2†§], Rouven L Ziegler[2†], Jan J Hirtz[3], Ernst Niebur[1,4]\*, Jan Pielage[2]\*

[1]Zanvyl Krieger Mind/Brain Institute, Johns Hopkins University, Baltimore, United States; [2]Division of Neurobiology and Zoology, Department of Biology, University of Kaiserslautern, Kaiserslautern, Germany; [3]Physiology of Neuronal Networks Group, Department of Biology, University of Kaiserslautern, Kaiserslautern, Germany; [4]Solomon Snyder Department of Neuroscience, Johns Hopkins University, Baltimore, United States

**\*For correspondence:**
niebur@jhu.edu (EN);
pielage@bio.uni-kl.de (JP)

[†]These authors contributed equally to this work

**Present address:** [‡]Yale MD-PhD Program, Yale School of Medicine, New Haven, United States; [§]German Center for Neurodegenerative Diseases (DZNE), Bonn, Germany

**Competing interest:** The authors declare that no competing interests exist.

**Abstract** The ability to associate neutral stimuli with valence information and to store these associations as memories forms the basis for decision making. To determine the underlying computational principles, we build a realistic computational model of a central decision module within the *Drosophila* mushroom body (MB), the fly's center for learning and memory. Our model combines the electron microscopy-based architecture of one MB output neuron (MBON-α3), the synaptic connectivity of its 948 presynaptic Kenyon cells (KCs), and its membrane properties obtained from patch-clamp recordings. We show that this neuron is electrotonically compact and that synaptic input corresponding to simulated odor input robustly drives its spiking behavior. Therefore, sparse innervation by KCs can efficiently control and modulate MBON activity in response to learning with minimal requirements on the specificity of synaptic localization. This architecture allows efficient storage of large numbers of memories using the flexible stochastic connectivity of the circuit.

## Editor's evaluation

In light of the ongoing emergence of volume electron microscopy connectomics, detailed morphologies at the nanometre scale for many neurons are now available, ready for functional and computational analysis. Building on these foundational resources, this work delivers compelling evidence that synaptic inputs onto the dendritic arborisations of readout neurons (MBONs) of the learning and memory system of the adult *Drosophila melanogaster* contribute with equal weight to the depolarization of the neuron, independently of their location on the arbor, a phenomenon known as synaptic democracy. These important findings establish the validity of computational models based on passive dendritic propagation for simulating fly brain circuits and highlight the differences between the much larger mammalian neurons that present active propagation strategies as part of their approach to synaptic democracy.

## Introduction

Operating successfully within a complex environment requires organisms to discriminate between a vast number of rewarding and stressful interactions. Learning to avoid potentially harmful interactions will promote survival of individual animals and requires long-term memory mechanisms. In recent years, significant progress has been made toward our understanding of the cellular and circuit

mechanisms underlying learning and memory. Despite these advances, at the computational level it remains largely unknown how the cellular and circuit architectures contribute to the efficient formation and storage of multiple memories.

Here, we use the *Drosophila melanogaster* mushroom body (MB) as a simple model system to investigate the computational principles underlying learning and memory in the context of decision making. A key advantage of the MB, the learning and memory center of the fly, is its relative simplicity in terms of the number and types of neurons compared to the mammalian brain. It has been demonstrated that the MB is required for the acquisition and recall of associative olfactory memories (*de Belle and Heisenberg, 1994*), and a wide variety of genetic tools are available to access and manipulate key cell types within the memory circuitry (*Owald et al., 2015b*; *Lai and Lee, 2006*).

Odor information is transmitted from the antennal lobe via projection neurons (PNs) to the Kenyon cells (KCs) of the MB. PNs are connected in a largely stochastic manner to the approximately 2000 KCs per brain hemisphere, resulting in a unique KC odor representation in individual flies (*Litwin-Kumar et al., 2017*; *Aso et al., 2014a*; *Caron et al., 2013*; *Gruntman and Turner, 2013*; *Lin et al., 2007*; *Betkiewicz et al., 2020*; *Eichler et al., 2017*). The exception to this is a partial non-random connectivity of a population of food-responsive PNs (*Zheng et al., 2022*). The formation of olfactory memories within the MB circuit is assayed through experimental manipulations in which normally neutral odors are associated with either positive (reward learning) or negative (avoidance learning) valences (*Livingstone et al., 1984*).

Electrophysiological and calcium imaging studies demonstrated that any given odor activates approximately 3–9% of KCs (*Campbell et al., 2013*; *Honegger et al., 2011*; *Turner et al., 2008*; *Siegenthaler et al., 2019*). KCs then transmit this odor information to mushroom body output neurons (MBONs) that subdivide the MB lobes into distinct structural and functional modules by virtue of their dendritic arborizations (*Owald et al., 2015a*; *Aso et al., 2014b*; *Hige et al., 2015a*; *Cognigni et al., 2018*). During associative olfactory conditioning, the behavioral response to any given odor can be altered to encode positive (approach) or negative (avoidance) behavior. The valence of the conditioned response is encoded by the activation of dopaminergic neurons (DANs) that selectively innervate the structural MB modules defined by the MBON dendrites (*Aso et al., 2014a*; *Saumweber et al., 2018*).

DANs of the PPL1 cluster preferentially provide input to the vertical lobe of the MB and are activated during negative reinforcement, while DANs of the PAM cluster are activated during positive reinforcement and modulate activity in the horizontal lobe of the MB (*Handler et al., 2019*; *Aso and Rubin, 2016*; *Tomchik and Davis, 2009*; *Séjourné et al., 2011*; *Cohn et al., 2015*; *Burke et al., 2012*; *Rohwedder et al., 2016*; *Saumweber et al., 2018*; *Selcho et al., 2009*). The majority of MBONs of the vertical lobe utilize acetylcholine as a neurotransmitter, and artificial activation of these MBONs often mediates approach behavior (*Aso et al., 2014b*; *Eschbach et al., 2021*). In contrast, the majority of MBONs of the horizontal lobe release glutamate as a neurotransmitter, and these neurons mediate avoidance behavior (*Aso et al., 2014b*; *Eschbach et al., 2021*).

In vivo calcium imaging and electrophysiological experiments demonstrated that the activity of individual MBONs can be altered by conditioning paradigms (*Hige et al., 2015a*; *Aso and Rubin, 2016*; *Plaçais et al., 2013*; *Owald et al., 2015a*; *Jacob and Waddell, 2020*; *Perisse et al., 2016*; *Felsenberg et al., 2018*). This modulation of MBON activity depends on the pairing of DAN and KC activity and results in either an enhancement or depression of the strength of the KC>MBON connection (*Handler et al., 2019*; *Owald et al., 2015a*; *Cohn et al., 2015*; *Burke et al., 2012*). These changes in KC>MBON connection strength occur in either the horizontal or the vertical lobe depending on the administered stimulus-reward contingencies. As a consequence, the resulting net activity of all MBONs is shifted to either approach or avoidance behavior, and the behavioral response toward the conditioned stimulus during memory recall is changed (*Handler et al., 2019*; *Aso and Rubin, 2016*; *Cohn et al., 2015*; *Burke et al., 2012*). Evidence for this MBON balance model (*Cognigni et al., 2018*) has been provided for short-term memory (*Aso et al., 2014a*; *Hige et al., 2015a*; *Perisse et al., 2016*; *Cohn et al., 2015*) but to what extent, if any, such a model applies to long-term memory remains unclear. Recently, genetic tools have been created that enable genetic tagging of long-term memory engram cells in the MB (*Miyashita et al., 2018*; *Siegenthaler et al., 2019*). These studies indicate that modulation of a relatively small number of KCs is sufficient to significantly alter memory recall.

Here, we generate a realistic computational model of a core KC>MBON module that is essential for the incorporation of memories to determine the cellular basis for decision making. Prior studies (*Gouwens and Wilson, 2009*) used 'synthetic' (randomly generated) models of dendritic trees of *Drosophila* projection neurons to gain first insights into the computational processes of central neurons. Using linear cable theory, they demonstrated that voltage differences are much smaller within the dendritic trees than between the dendrites and the soma. Using a similar approach, Scheffer and coworkers simulated voltage distributions in the dendritic tuft of another *Drosophila* neuron (EPG) but used its measured morphology, rather than random geometries (*Scheffer et al., 2020*). Here, we go beyond these studies by combining precise structural data including the full synaptic connectivity from the electron microscopy-based synaptic connectome (*Takemura et al., 2017*) of the *Drosophila* MB with functional (electrophysiological) data of an individual MBON. This approach allows us to determine the impact of populations of KCs on MBON activity and to resolve the potential computational mechanisms underlying memory encoding. We focus on MBON-α3 (MBON-14 in the terminology of *Aso et al., 2014a*), an MBON at the tip of the α-lobe that is relevant for long-term memory (*Aso et al., 2014b*; *Plaçais et al., 2013*). We first determine the physiological parameters of MBON-α3 by patch-clamp electrophysiology. We then use these parameters to generate an anatomically and physiologically realistic in silico model of MBON-α3 and its KC inputs. Using a variety of computational simulations, we provide evidence that MBON-α3 is electrotonically compact and perfectly suited to incorporate a substantial number of memories based on largely stochastic KC inputs.

## Results

### Electrophysiological characterization of MBON-α3

MBON-α3 is particularly suitable for computational modeling due to the availability of its complete dendritic reconstruction through electron microscopy (*Takemura et al., 2017*). In addition, we can access MBON-α3 genetically by using a specific Gal4 line (MB082C-Gal4) to label MBON-α3 by the expression of membrane-bound GFP (10xUAS-IVS-mCD8::GFP; *Figure 1A*). In immunohistochemical co-labelings with the active zone marker Bruchpilot (Brp), we were able to visualize the dendritic input sites, the axonal projections and the cell soma of MBON-α3 (*Figure 1A*, arrow). We used this labeling to perform fluorescence-guided electrophysiological whole-cell patch-clamp recordings of this neuron ex vivo (*Figure 1C and D*). We carried out standard patch-clamp protocols in five cells of independent preparations and recorded all essential physiological parameters (*Table 1*). It is of note that we cannot discriminate between the two different MBON-α3 cells that are marked by the split Gal4 line and that are almost identical with regard to their dendritic tree position and synaptic input number (*Takemura et al., 2017*). First, we performed a continuous current-clamp protocol over $60s$ at a sampling rate of $50kHz$ which allowed us to determine the average resting membrane potential as $-56.7 \pm 2.0mV$. This value for the resting membrane potential is in line with prior measurements of MBONs in vivo and of other central neurons in *Drosophila* (*Hige et al., 2015a*; *Wilson et al., 2004*; *Gu and O'Dowd, 2006*; *Groschner et al., 2022*). In these recordings, we observed spontaneous firing activity of MBON-α3 with an average frequency of $12.1Hz$. Next, we performed a current-clamp de- and hyper-polarization protocol where we recorded $400ms$ trials at a sampling rate of $50kHz$. Each trial started with $0pA$ for $10ms$, followed by a current injection of $-26pA$ to $32pA$, with $2pA$ increments for $400ms$. These protocols allowed us to evoke action potentials and to determine the resulting changes in membrane potential. An example displaying selected traces of a representative recording is shown in *Figure 1F*. Two additional examples are illustrated in *Figure 1—figure supplement 1*. From this data, we determined the average membrane time constant as $\tau_m = 16.06 \pm 2.3ms$. We then performed a voltage-clamp step protocol without compensating for the series resistance. In this experiment, we recorded at $20kHz$ while applying a $5mV$ voltage step for $100ms$. This allowed us to calculate the average membrane resistance as $R_m = 926 \pm 55M\Omega$, a rather high membrane resistance when compared to similarly sized neurons (*Gouwens and Wilson, 2009*). MBON-α3 is thus highly excitable, with small input currents being sufficient to significantly alter its membrane potential.

The amplitude of action potentials was rather small $(4.3 \pm 0.029mV)$ but similar to previously reported values for different MBONs (*Hige et al., 2015a*). The small amplitudes are likely a result of the unipolar morphology of the neuron with a long neurite connecting the dendritic input region to the soma (see below). As a result, signal propagation may partially bypass the soma (*Figure 1A, B and*

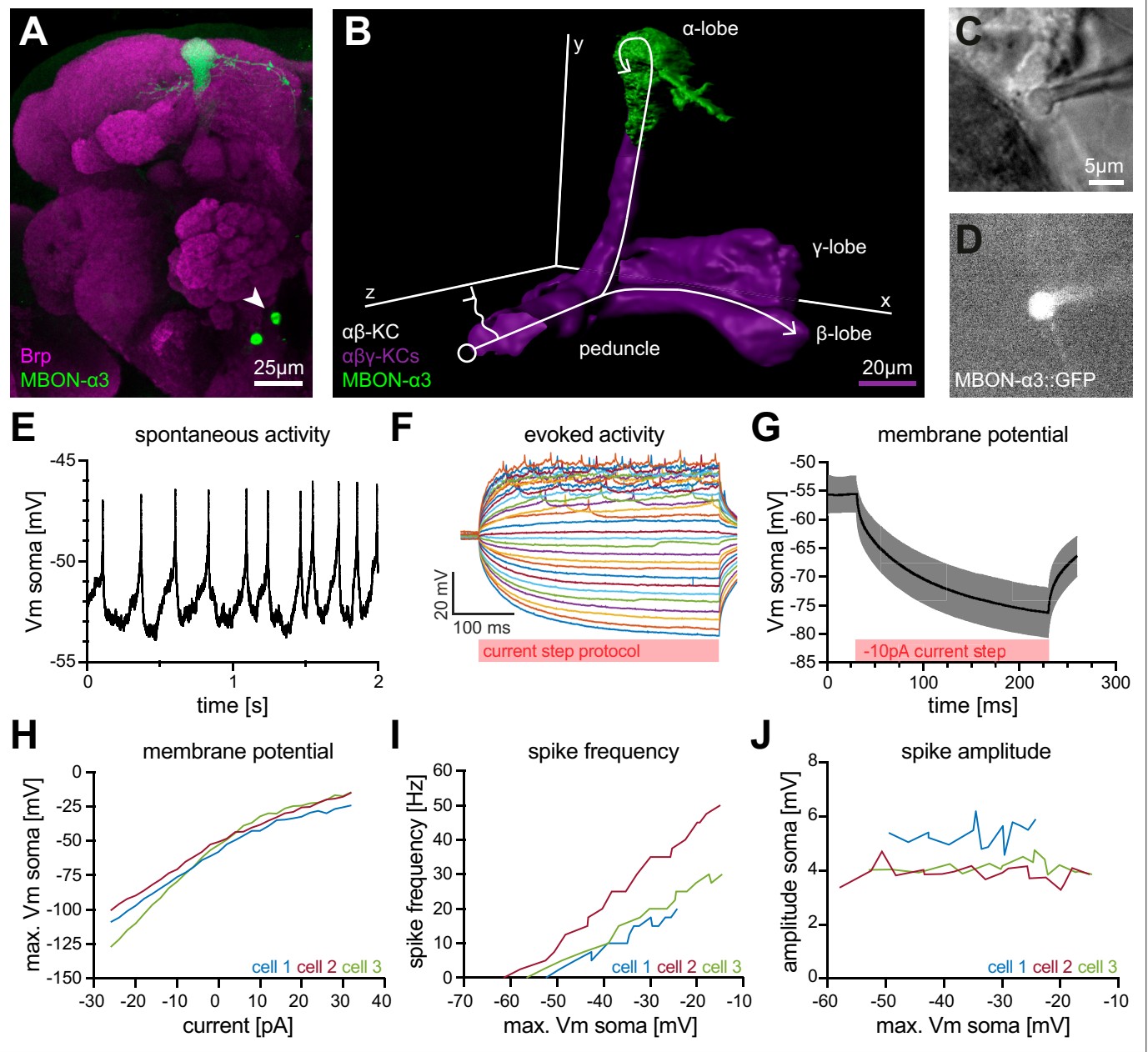

**Figure 1.** Electrophysiological characterization of the MBON-α3 neuron. (**A**) Immunohistochemical analysis of the MBON-α3 (green, marked by GFP expression) reveals the unique morphology of the MBON with its dendritic tree at the tip of the α-lobe and the cell soma (arrowhead) at a ventro-medial position below the antennal lobes. The brain architecture is revealed by a co-staining with the synaptic active zone marker Bruchpilot (Brp, magenta). (**B**) Artificial superimposition of a partial rendering of MBON-α3 (green) and of $\alpha\beta\gamma$-KCs (magenta) to illustrate KC>MBON connectivity. A simplified schematic $\alpha\beta$-KC projection is illustrated in white. (**C**) Transmitted light image of an MBON-α3 cell body attached to a patch pipette. (**D**) Visualisation of the GFP-expression of MBON-α3 that was used to identify the cell via fluorescence microscopy. The patch pipette tip is attached to the soma (right side). The scale bar in C applies to C and D. (**E**) Recording of spontaneous activity of an MBON-α3 neuron in current clamp without current injection. (**F**) Recording of evoked neuronal activity of an MBON-α3 neuron with step-wise increasing injection of $400ms$ current pulses. Pulses start at $-26pA$ (bottom) with increasing $2pA$ steps and end at $+32pA$ (top). (**G**) Mean trace of the induced membrane polarization resulting from a $200ms$ long current injection of $-10pA$. 105 trials from three different cells (35 each) were averaged. Gray shading indicates the SD. (**H**) Relative change of membrane potential after current injection. Vm is equal to the maximum depolarization of the membrane. (**I**) Correlation between the frequency of action potential firing and changes in membrane potential. (**J**) Relative change of spike amplitude after current injections. Spike amplitude represents the peak of the largest action potential minus baseline. Also see *Figure 1—figure supplement 1* and *Figure 1—figure supplement 2*.

The online version of this article includes the following figure supplement(s) for figure 1:

**Figure supplement 1.** Additional information about the electrophysiology of MBON-α3.

**Figure supplement 2.** Electrophysiological properties of MBON-α3.

**Table 1.** Summary of passive membrane properties of MBON-α3, calculated from electrophysiological measurements from five neurons.

From left to right: resting membrane potential ($V_m$) in $mV$, membrane time constant ($\tau_m$) in $ms$, capacitance of the membrane ($C_m$) in $pF$, specific capacitance ($C_{spec}$) in $\mu F/cm^2$ and passive membrane conductance ($C_{pass}$) in $S/cm^2$. *Table 1—source data 1* Summary of passive membrane properties of MBON-α3, calculated from electrophysiological measurements from four neurons marked by cytoplasmic EGFP.

| Sample | $Vm[mV]$ | $\tau_m[ms]$ | $C_m[pF]$ | $C_{spec}[\mu F/cm^2]$ | $C_{pass}[S/cm^2]$ |
|---|---|---|---|---|---|
| Cell 1 | −59.2 | 13.54 | 12.35 | 0.200 | $1.87 \times 10^{-5}$ |
| Cell 2 | −52.3 | 14.46 | 20.79 | 0.337 | $2.12 \times 10^{-5}$ |
| Cell 3 | −56.2 | 24.58 | 21.60 | 0.350 | $1.48 \times 10^{-5}$ |
| Cell 4 | −52.8 | 15.50 | 15.83 | 0.257 | $1.66 \times 10^{-5}$ |
| Cell 5 | −63.1 | 12.20 | 13.21 | 0.214 | $1.76 \times 10^{-5}$ |
| Mean | −56.7 | 16.06 | 16.76 | 0.272 | $1.78 \times 10^{-5}$ |
| SD | 4.5 | 4.92 | 4.26 | 0.069 | $0.24 \times 10^{-5}$ |
| SEM | 2.0 | 2.20 | 1.90 | 0.031 | $0.11 \times 10^{-5}$ |

The online version of this article includes the following source data for table 1:

**Source data 1.** Summary of the passive membrane properties of MBON-α3, calculated from electrophysiological measurements from four neurons marked by cytoplasmic EGFP.

*F*; *Gouwens and Wilson, 2009*). We calculated the neuron's membrane capacitance ($C_m = \tau_m/Rm$) as $C_m = 16.76 \pm 1.90 pF$, which classifies MBON-α3 as a mid-sized neuron.

To enable the calibration of the in silico model, we then recorded 35 trials per cell in a current-clamp injection protocol with injections of $10pA$ current pulses for $200ms$ at a sampling rate of $20kHz$. We averaged 105 traces from three selected example cells to a standard curve, describing the membrane kinetics of MBON-α3 in response to current injections (*Figure 1G*). To further characterize MBON-α3, we analyzed the resulting data from the current-clamp de- and hyper-polarization protocols in the three example cells (*Figure 1F* and *Figure 1—figure supplement 1*). We recorded the absolute deflection, the action potential firing frequency, and the amplitude of action potentials for all individual cells. In all three cells, we observed that the absolute deflection of the membrane potential was proportional to the increasing current injections (*Figure 1H*). The action potential firing frequency increased gradually with increasing membrane potential deflections (*Figure 1I*), with no significant change in action potential amplitudes (*Figure 1J*). The same effect was observed with an increasing current injection (*Figure 1—figure supplement 1*). This is consistent with the general idea that action potentials are "all or nothing" events, and that MBON-α3 is a spike-frequency adapting neuron constistent with prior observations of other MBONs (*Hige et al., 2015a*). We repeated these recordings with a cytosolic EGFP as an independent and non-membrane associated marker and observed consistent results (*Figure 1—figure supplement 2* and *Table 1—source data 1*). We next used these data to generate an in silico model based on realistic passive membrane properties.

## Constructing an in silico model of MBON-α3

For our in silico model of MBON-α3, we used the recently published electron-microscopy-based connectome of the *Drosophila* mushroom body (*Takemura et al., 2017*). This dataset includes the precise morphological parameters of the entire dendritic tree including the precise location data for all 12,770 synaptic connections from the 948 innervating KCs. It is important to note that this dataset represents a single fly and the EM-based reconstruction was not able to faithfully assign all synaptic connections (potentially missing 10% of synapses; *Takemura et al., 2017*). The comparison with the hemibrain data set (*Scheffer et al., 2020*) that provides a second independent reconstruction of the MBON from a different fly showed very few differences in KC to MBON-α3 connectivity. Thus, despite potential shortcomings, this dataset represents the most accurate template to reconstruct the dendritic tree and connectivity of the MBON. To build a complete morphological reconstruction of

MBON-α3, we included the axonal reconstructions from the hemibrain dataset (*Scheffer et al., 2020*) and determined the length of the connection between the soma and the dendritic region via confocal light microscopy. We incorporated these data together with all other values into the NEURON simulation environment (*Hines and Carnevale, 1997*; *Figure 2A and C* and *Figure 2—figure supplement 1*).

Based on the published vector data (*Takemura et al., 2017*) we subdivided the dendritic tree into 4336 linear sections, of which 3121 are postsynaptic to one or more of the 12,770 synaptic contacts from 948 KCs (*Figure 2B*). We used linear cable theory (*Niebur, 2008*) to generate the in silico model of MBON-α3. Linear cable theory is the baseline model for neurites and the most rational approach in the absence of detailed information about potential nonlinear currents.

In previous simulations of *Drosophila* neurons, the kinetics of membrane polarisation as a response to current injections were adjusted to in vivo measurements through a fitting procedure (*Gouwens and Wilson, 2009*). We first defined the boundaries of the passive membrane parameters within NEURON's PRAXIS (principle axis) optimization algorithm and incorporated the experimentally defined passive membrane properties (*Table 1*). We then determined the biophysical parameters of the in silico neuron by current step injections of $-10pA$ for $200ms$ at the soma and by recording the neuronal voltage traces. This enabled us to fit the membrane parameters of our model neuron to the experimental data of the three example cells (*Figure 2E*). The biophysical parameters that resulted from the fitting procedure are provided in *Table 2*.

We obtained membrane kinetics that closely resemble ex vivo current injections of $-10pA$, with a mean squared error between model and measured data of $0.0481mV^2$. For comparison, we normalized both the modelled current injection (*Figure 2D*) and the average of the traces from the ex vivo recordings (*Figure 2E*) and overlayed the resulting traces in *Figure 2F*.

Note that the faster voltage change at the onset and offset of the current injection seen in the simulation is caused by the small size of the soma which results in a much faster 'local' somatic time constant than that of the whole cell (*Figure 2—figure supplement 2*). This is also visible in the average of the recorded traces (*Figure 2F*), but more obvious in individual (not trial-averaged) recorded traces (*Figure 2—figure supplement 2C*). Overall, these results indicate that we were able to generate a realistic in silico model of MBON-α3.

## Effect of KC>MBON synaptic inputs

As KCs are cholinergic (*Barnstedt et al., 2016*) we utilized previously determined parameters of cholinergic synapses to simulate synaptic input to MBON-α3. We used a reversal potential of $8.9mV$ and a synaptic time constant $\tau_s = 0.44ms$ as target values (*Su and O'Dowd, 2003*). Individual synaptic contacts were simulated as alpha functions ($g_{max}(t - t_i) \times \exp(-(t - t_i - \tau_s)/\tau_s)$) where $t_i$ is the time of the incoming spike and $g_{max} = 1.56 \times 10^{-11}S$, chosen to obtain response levels in agreement with the target values. The average number of synaptic contacts from a KC to this MBON is 13.47 (*Takemura et al., 2017*), Figure 4B. To determine the computational constraints of MBON-α3, we first stimulated each of the 3,121 dendritic sections that receive KC synaptic innervation using the described alpha-function conductance change. We then "recorded" (from our simulation) the resulting voltage excursions at the dendritic input site (*Figure 3A*, red), at the proximal neurite (putative action potential initiation site, green) and at the cell soma (blue). The comparative analysis of the responses at these three locations revealed two notable features: (1) voltage excursions directly within the dendrite are faster and substantially larger than in the proximal neurite and soma, and (2), the resulting voltages in the latter two compartments are much less variable than in the dendrite and distributed within a very small voltage range (*Figure 3D and E*). These results indicate that the morphological architecture and biophysical parameters of MBON-α3 promote a 'compactification', or 'democratization', of synaptic inputs, with all inputs resulting in similar voltage excursions at the proximal neurite and soma, regardless of the strength of the initial dendritic voltage or the position on the dendritic tree.

To a first approximation, compactification can be understood from a simple analytical estimate of the space constant of this neuron. It is obtained by assuming a non-branching cylinder with the mean diameter of the dendritic segments, $0.29\mu m$, as obtained from the morphological data from *Takemura et al., 2017*. Taking the values for the resistivity of the cytoplasm ($R_a = 85\Omega cm$) and the transmembrane conductance ($g_{pas} = 9.4 \times 10^{-6}S/cm^2$) from *Table 2*, the characteristic length over which excitations decay can then be computed by linear cable theory (*Niebur, 2008*) as $\lambda \approx 1,300\mu m$.

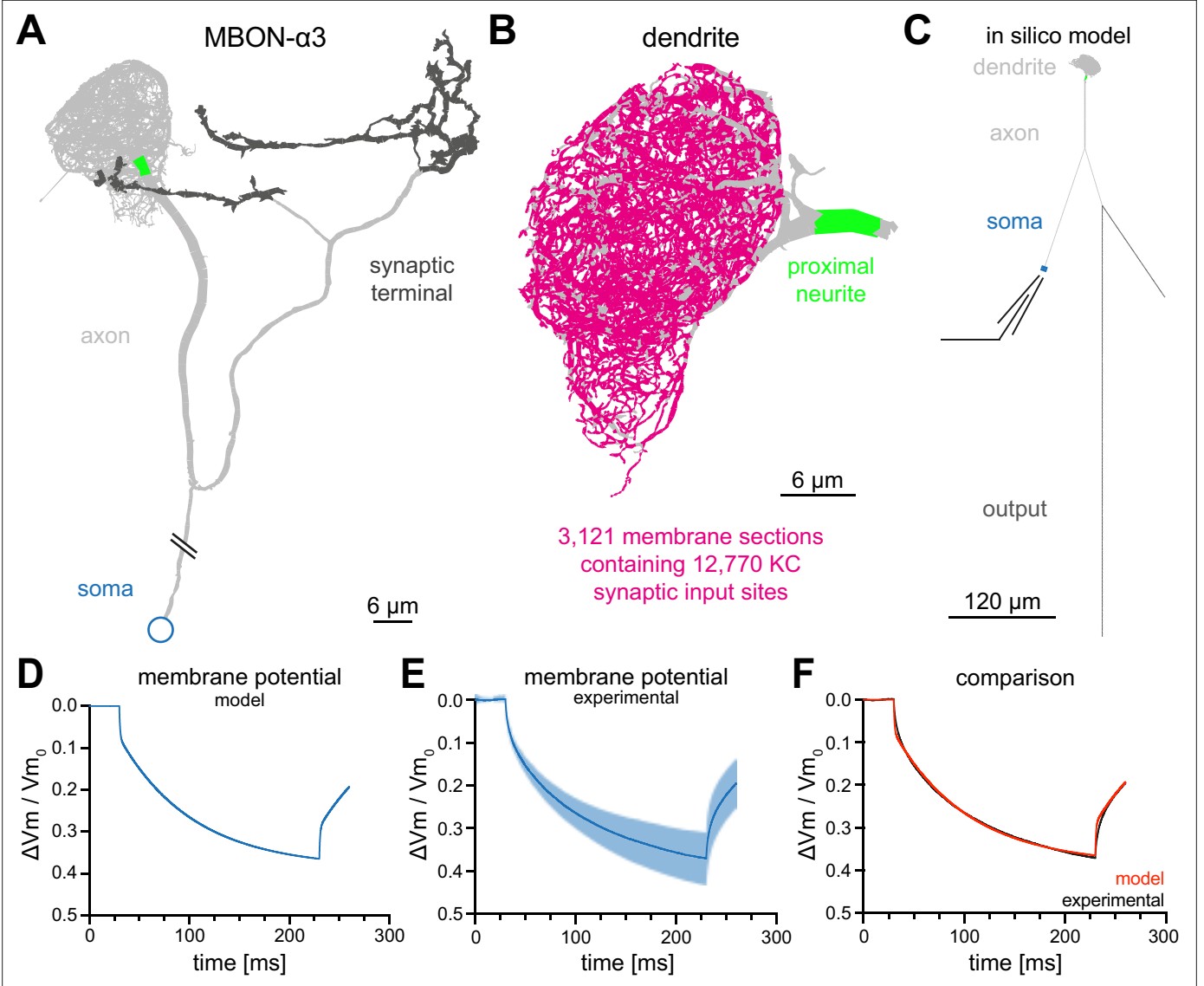

**Figure 2.** Construction of a computational model of MBON-α3. (**A**) Electron microscopy based reconstruction of MBON-α3. Data was obtained from NeuPrint (*Takemura et al., 2017*) and visualized with NEURON (*Hines and Carnevale, 1997*). Data set MBON14 (ID 54977) was used for the dendritic architecture (**A–C**) and MBON14(a3)R (ID 300972942) was used to model the axon (light gray) and synaptic terminal (dark gray, **A, C**). The proximal neurite (bright green) was defined as the proximal axonal region next to the dendritic tree. This is the presumed site of action potential generation. Connectivity to the soma (blue) is included for illustration of the overall morphology and is not drawn to scale. (**B**) A magnified side view of the dendritic tree of MBON-α3. Within the dendritic tree a total of 4336 individual membrane sections were defined (light gray area). The 3121 membrane sections that contain one or more of the 12,770 synaptic input sites from the 948 innervating KCs are highlighted in magenta. (**C**) Simplified in silico model of MBON-α3 highlighting the size of the individual neuronal segments. We simulated recordings at the proximal neurite (specified as the potential site of action potential generation based on morphological parameters; this area is still included in the dendritic tree reconstruction of *Takemura et al., 2017*) or at the soma. Values for the different sections are provided in *Figure 2—figure supplement 1*, *Figure 2—source data 1*. (**D**) Normalized trace of a simulated membrane polarization after injection of a $200ms$ long square-pulse current of $-10pA$ at the soma. (**E**) Normalized and averaged experimental traces (blue) with standard deviation (light blue) of our measured depolarization. (**F**) Comparison of a normalized induced depolarization from the model (red) and from the experimental approach (black). The model was fitted within the NEURON environment to the measured normalized mean traces with a mean squared error between model and measured data of $0.048mV^2$. Also see *Figure 2—figure supplement 1*, *Figure 2—figure supplement 2* and *Figure 2—source data 1*.

The online version of this article includes the following source data and figure supplement(s) for figure 2:

**Source data 1.** Numerical values of the individual sections of the neuron utilized for the computational model.

**Figure supplement 1.** Overview of the axonal sections and soma in the in silico model.

**Figure supplement 2.** Origin of the fast voltage change followed by slower change subsequent to current steps, visible in *Figure 2D, E and F*.

**Table 2.** Electrophysiological values applied for the in silico model of MBON-α3-A. Parameter values were obtained from literature or were based on the fitting to our electrophysiological data. The first column shows the parameters, with the names used in the NEURON environment in parentheses. The maximal conductance was determined to achieve the target MBON depolarization from monosynaptic KC innervation.

**Electrophysiological Properties**

| Variable | Description | Fitted | Literature |
|---|---|---|---|
| $R_a$ (Ra) | Cytoplasm resistivity [$\Omega \times cm$] | 85.41 | 30–400 (**Borst and Haag, 1996**; **Gouwens and Wilson, 2009**) |
| $C_m$ (cm) | Specific membrane capacitance [$\mu F/cm^2$] | 0.6961 | 0.6–2.6 (**Borst and Haag, 1996**) |
| $g_{pas}$ (pas.g) | Passive membrane conductance [$S/cm^2$] | $9.399 \times 10^{-6}$ | $3.8 \times 10^{-5} - 1.2 \times 10^{-4}$ (**Cassenaer and Laurent, 2007**) |
| $e_{pas}$ (pas.e) | Leak reversal potential [$mV$] | –55.64 | –60 (**Berger and Crook, 2015**) |

**Synaptic Parameters**

| | | | |
|---|---|---|---|
| $\tau_s$ (tau) | Time to max. conductance [$ms$] | N.A. | 0.44 (**Su and O'Dowd, 2003**) |
| E | Synaptic current reversal potential [$mV$] | N.A. | 8.9 (**Su and O'Dowd, 2003**) |
| $G_{max}$ (gmax) | Maximal conductance [$\mu mho$] | N.A. | $1.5627 \times 10^{-5}$ (**Hige et al., 2015b**) |

This is about twice the maximal path length between any two segments of the cell, see **Figure 2C**. Using the average diameter of all neuronal compartments, both in the dendritic tree and in the other components of the neuron as listed in **Figure 2—figure supplement 1**, increases $\lambda$ by about 10% because the non-dendritic segments have larger diameters. In either case, synaptic input even at distal locations on the dendritic tree can influence somatic voltage deflections nearly as strongly as more proximal synapses, see **Figure 3E**.

To understand additional influences of cell morphology for the input computation, we analyzed the relation between synaptic position and local dendritic volume and the resulting voltages in the dendrite, proximal neurite, and soma. Representing the distance between dendritic inputs and soma (**Figure 3G**) by a color code, we find that, as in many tree-like structures, there is a systematic tendency that more distal segments (more distant from the soma) have a smaller volume and diameter, **Figure 3H**. The same figure also shows that the simulated voltage excursions are many times larger in distal, small segments than in proximal, large segments, over a range from $\approx 0.03mV$ to $\approx 1mV$. This is expected: placing an electrical charge (integrated synaptic current) on a small capacitor (low-volume segment with small surface area) results in a larger voltage excursion than placing the same charge on a large capacitor (large segment). It is highly remarkable, however, that the morphology of the MBON-α3 dendritic tree seems finely tuned, with the goal of maximizing the equality of the functional weight of all synapses. Specifically, the decrease of the voltage excursion in the soma caused by the injection of a current is nearly perfectly (within less than $0.5\mu V$) compensated by the increase of the local dendritic voltage due to the smaller volume of distal segments, see **Figure 3I**. The approximately 30-fold difference of local dendritic voltages ($\approx 0.03mV$ to $\approx 1mV$) is nearly exactly compensated, such that activation of each synapse has exactly the same effect at the soma, within less than one $\mu V$.

We next analyzed the impact of individual KCs on MBON-α3. We simulated the activation of all synapses of each individual KC that innervates MBON-α3, using the synaptic parameters described above, and we recorded the simulated voltage excursions in the soma, **Figure 4A**. The number of

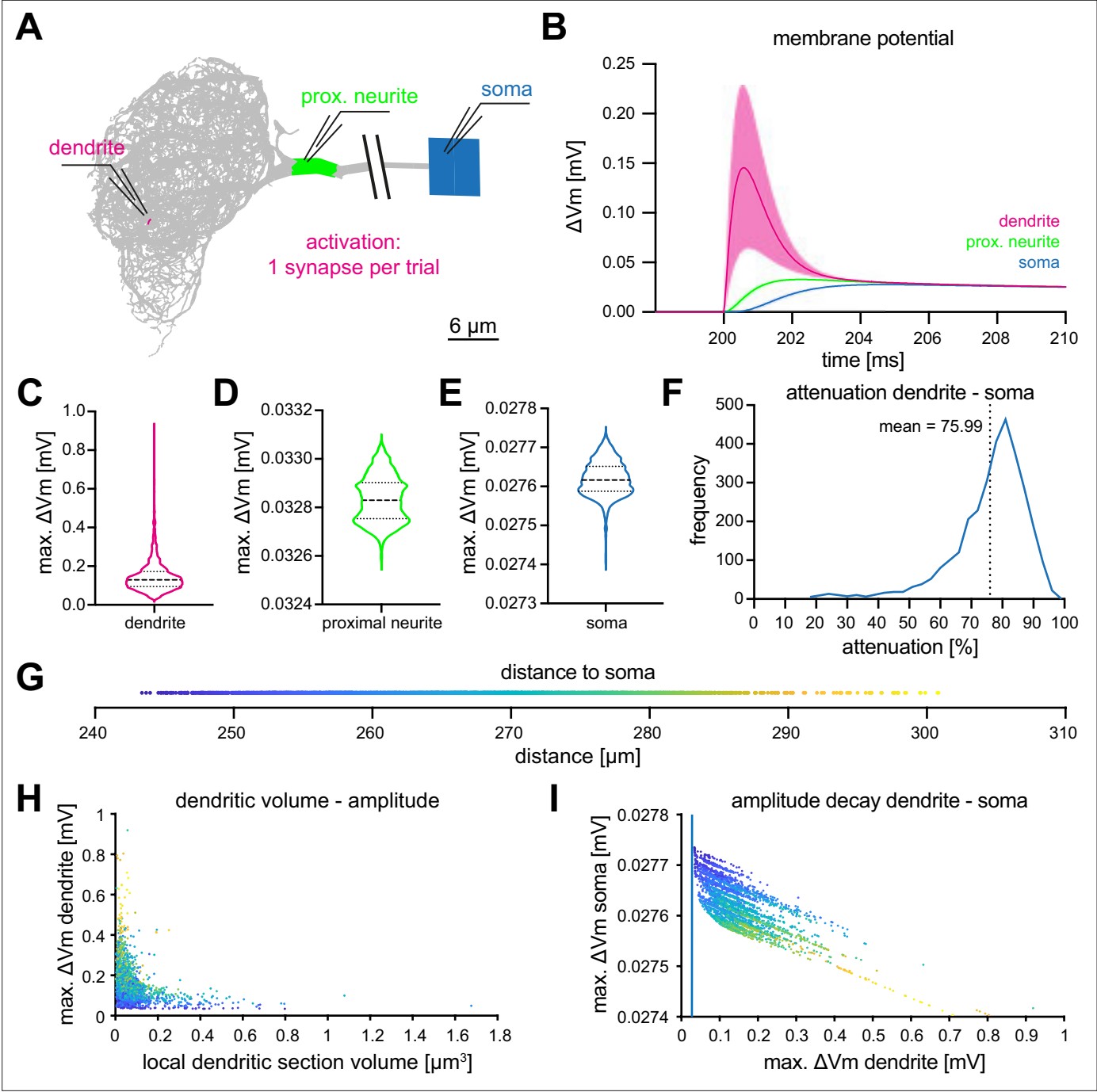

**Figure 3.** Computational characterization of MBON-α3. (**A**) Simulated stimulation of individual synaptic input sites. Voltage deflections of synaptic currents were analyzed locally at the dendritic segment (red, next to electrode tip), the proximal neurite (green) and the soma (blue). The soma is depicted as a square that reflects its size in the NEURON environment. Color code also applies to B-F. (**B**) Resulting mean voltages (lines) and standard deviations (shading) from synaptic activations in 3121 individual stimulus locations (dendrite), and at the proximal neurite and soma. (**C–E**) Violin plots of the simulated maximal amplitudes at the dendritic section (**C**), the proximal neurite (**D**) and the soma (**E**) in response to synaptic activations. Dotted lines represents the quartiles and dashed lines the mean. Proportions of the violin body represent the distribution of individual data points. Note the difference in scale and compactification of the amplitudes in the proximal neurite and soma. (**F**) Distribution of the percentage of voltage attenuation recorded at the soma. (**G**) Distribution of distance between all individual dendritic sections and the soma. This color code is applied in H and I. (**H**) The elicited local dendritic depolarization is plotted as a function of the local dendritic section volume. (**I**) The soma amplitude is plotted as a function of the amplitude at individual dendritic segments. Note different scales between abscissa and ordinate; the blue line represents identity. Also see *Figure 3—figure supplement 1*.

*Figure 3 continued on next page*

*Figure 3 continued*

The online version of this article includes the following figure supplement(s) for figure 3:

**Figure supplement 1.** Additional information about voltage attenuation at the proximal neurite and soma.

synapses varied between KCs, ranging from 1 to 27 (plus one outlier with 38 synaptic contacts), with a mean of 13.47 synapses per KC (*Figure 4B*).

In these simulations, we observed a wide range of voltage excursions (note the outlier in *Figure 4B, C, D and E*). The neuron is firmly in a small-signal operation mode, as we observed a highly linear relation between the voltage excursion and the number of synapses per KC, *Figure 4E*. Activating a single KC leads to a voltage excursion at the soma with a mean of $0.37mV$, *Figure 4D*. To further analyze this linear relationship between synapse number and somatic voltage excursion, we analyzed all KCs with exactly 12, 13, or 14 synaptic inputs to MBON-α3. This analysis revealed a highly stereotypical depolarization at the soma for all these KCs, despite wide variations in the position of the individual synaptic contacts (*Figure 4E and F*). Activation of 13 randomly selected synapses in 1000 independent trials resulted in similar voltage excursions as observed for the KCs with 13 synaptic connections. Interestingly, in these simulations the responses showed less variability compared to the activation of individual KCs with synapses at anatomically observed locations (*Figure 4F*). Together, these results provide further support for the electrotonical compactness of MBON-α3.

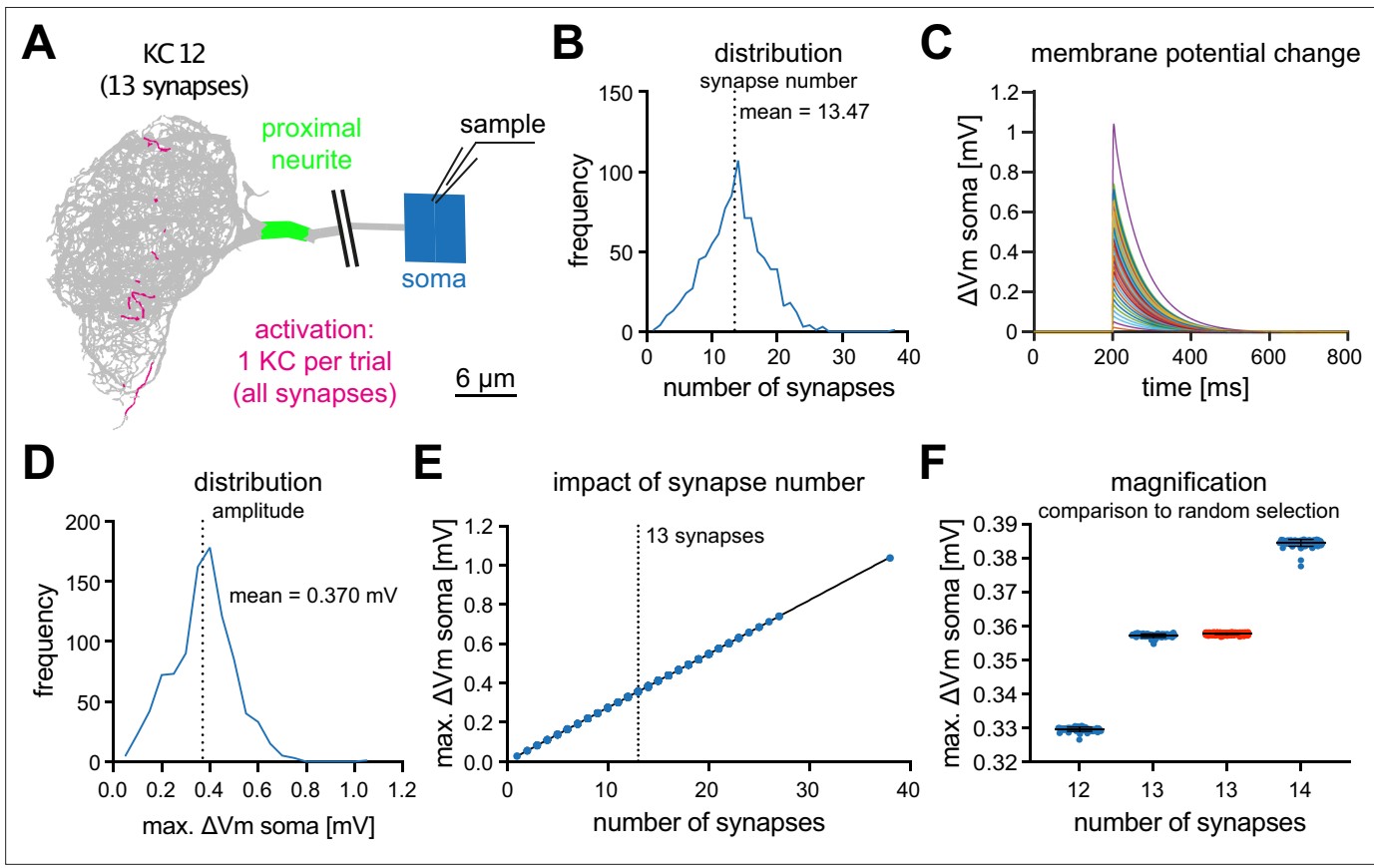

**Figure 4.** MBON-α3 is electrotonically compact. (**A**) Simulated activation of all synapses from all 948 individual KCs innervating MBON-α3. A representative example of a KC (KC12) with 13 input synapses to MBON-α3 is shown. (**B**) Histogram of the distribution of the number of synapses per KC. (**C**) Membrane potential traces from simulated activation of all 948 KCs. (**D**) Distribution of the elicited amplitudes evoked by the individual activations of all KCs. (**E**) Correlation between the somatic amplitude and the number of activated synapses in the trials. (**F**) Blue: Distribution of the somatic voltage change evoked by the activation of all KCs with exactly 12, 13, or 14 input synapses. Each dot represents one simulated KC. Bars represent mean and SD. Red: Same, but for activation of 13 random synapses per simulation in 1000 independent trials.

## Physiological and tuned activation of MBON-α3

A given odor activates only a small fraction of KCs reliably, reported as about 5% in some studies (*Honegger et al., 2011*; *Siegenthaler et al., 2019*) and 6% in others (*Turner et al., 2008*; *Campbell et al., 2013*). This one-percentage point difference may well be caused by differences in the experimental preparations. However, in absolute terms a change from 5% to 6% in the same system represents a 20% increase. To understand the physiological effect of differences in the number of activated KCs of this size, we first established a baseline condition by simulating voltage excursions in the soma resulting from the simultaneous activation of sets of 50 KCs (corresponding to 5%), randomly selected from the 948 KCs innervating MBON-α3 and addressed increases in KC number afterwards.

In 1000 independent activation trials (*Figure 5A*), we observed highly stereotypical activation patterns, with a mean somatic depolarization of $15.24 mV$ and only small variations in the time course (*Figure 5B and C* (blue)). These results are consistent with an activation of 50 KCs being sufficient to induce changes in the firing frequency of MBON-α3 (*Figure 1H, I*). The fact that all simulations resulted in a significant depolarization consistent with action potential generation or alterations of the firing frequency together with the small range of variation within this dataset, provides support for the current model that odor encoding in KCs is, at least to a large extent, random (stochastic; *Caron et al., 2013*; *Zheng et al., 2022*; *Li et al., 2020*; *Eichler et al., 2017*).

Based on in vivo data, two different modes of plasticity may act at the level of the KC-MBON module to alter MBON output to conditioned odors. In vivo imaging of short-term memory formation demonstrated alterations of KC input strength (*Owald and Waddell, 2015*; *Hige et al., 2015a*; *Perisse et al., 2016*; *Plaçais et al., 2013*; *Pai et al., 2013*; *Séjourné et al., 2011*; *Jacob and Waddell, 2020*) while recent observations of long-term memory demonstrated a specific change (increase) in the number of KCs representing the conditioned odor (*Delestro et al., 2020*; *Shyu et al., 2019*; *Baltruschat et al., 2021*). To directly address the relative impact of these two modulations on MBON depolarization, we repeated the same simulation but now modified input strength by either altering activated KC numbers or by modulating the synaptic gain. To enable a direct comparison, we modulated each factor by 25%. First, we increased the number of activated KCs per set by 25%, from 50 to 63. This lead to a mean somatic depolarization of $18 mV$, a significant increase by 19% (*Figure 5B and D*; dark blue). Decreasing the number of activated KCs by 25%, from 50 to 38, lead to a mean somatic depolarization of $12 mV$, a significant decrease by 20% (*Figure 5B and D*; light blue). Next, we either increased or decreased synaptic strength within sets by 25%. Increase lead to a mean somatic depolarization of $18 mV$, a significant increase by 19% (*Figure 5C and E*; purple) while decrease lead to a mean somatic depolarization of $12 mV$, a significant decrease by 21% (*Figure 5C and E*; red). Averaged over all conditions, a change of input of 25% lead to a change of 20±1% in membrane depolarization.

Even though the integrated synaptic conductance was changed by the same amount by our manipulation of either synaptic strength or number of activated KCs, we observed small but significant differences in the resultant mean somatic amplitudes (*Figure 5F and G*). Finally, we combined both manipulations by compensating an increase in KC number with a corresponding decrease in synaptic strength and vice versa. Both manipulations resulted, on average, in a significant decrease of MBON membrane depolarization (*Figure 5*). An overview of the performed simulations and numerical results is provided in *Table 3*.

Our simulations revealed a nearly linear relationship between the activated number of synapses and the measured amplitude in the soma regardless of synaptic strength (*Figure 5I*). Interestingly, we observed some differences in the slope of the responses between the different tuning modalities (*Figure 5J, K and L*), indicating that these modulations are not completely equivalent.

## Discussion

For our understanding of the computational principles underlying learning and memory, it is essential to determine the intrinsic contributions of neuronal circuit architecture. Associative olfactory memory formation in *Drosophila* provides an excellent model system to investigate such circuit motifs, as a large number of different odors can be associated with either approach or rejection behavior through the formation of both short- and long-term memory within the MB circuitry (*Aso and Rubin, 2016*). In contrast to most axon guidance processes in *Drosophila* that are essentially identical in all wild-type individuals, processing of odor information in the MB largely, but not exclusively, depends on the

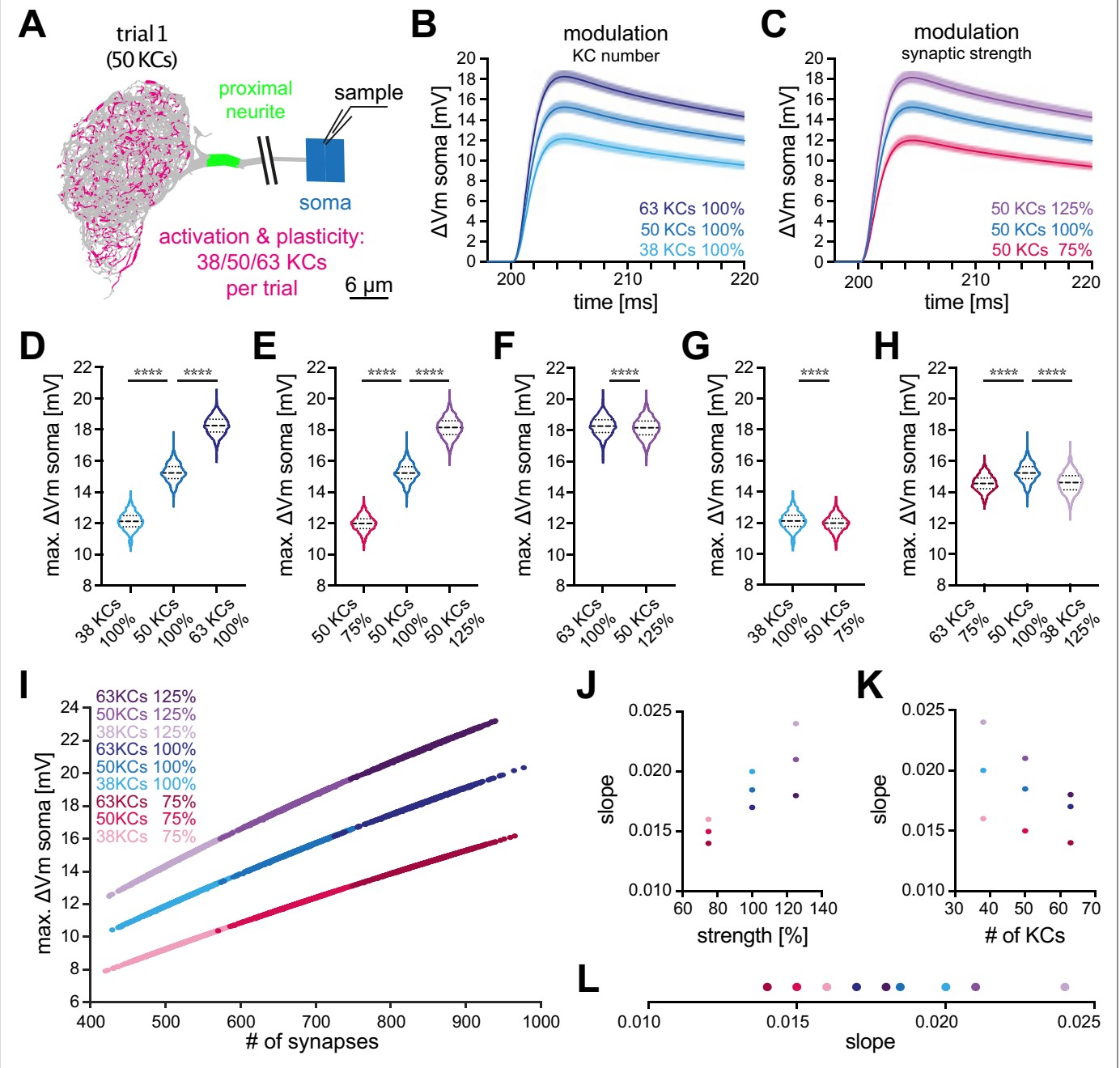

**Figure 5.** Effect of KC recruitment and synaptic plasticity on MBON-α3 responses. (**A**) Schematic of the simulated recording paradigm. We modeled the activation of ≈5% of random combinations of identified KCs, that is we activated the anatomically correct synapse locations of 50 randomly selected KCs to mimic activation by an odor. The panel shows the distribution of activated synapses for one of these trials. We then varied the number of activated KCs by ±25% around its mean of 50, that is 38, 50, and 63 KCs, and we varied synaptic strength ±25% around its original value, that is 75%, 100%, and 125%. (**B**) Mean somatic depolarizations after activating 1000 different sets of either 38 (light blue), 50 (blue), or 63 (dark blue) KCs, all at standard synaptic strength (100%). Shades represent the standard deviations. (**C**) Mean somatic depolarizations after activating 1000 different sets of 50 KCs at either 75% (red), 100% (blue), or 125% (purple) synaptic strength. Shades represent the standard deviations. (**D, E**) Violin plots of the relative amplitudes evoked by the different KC sets from (**B, C**). (**F–G**) Violin plots for the comparison of the amplitudes between the different plasticity paradigms. (**F**) 63 KCs at 100% *vs* 50 KCs at 125% synaptic strength. (**G**) 38 KCs at 100% *vs* 50 KCs at 75% synaptic strength. Plots in D-G use the color code from (**B**) and (**C**). (**H**) 63 KCs at 75% (dark red) *vs.* 50 KCs at 100% (blue) and 38 KCs at 125% (light purple) synaptic strength. See *Figure 3* for symbols in violin plots. (**I**) Relation between somatic depolarization and the number and strength of activated synapses. Illustrated are all tested conditions for synaptic plasticity and KC recruitment. Stimulation paradigms are color coded as shown. (**J**) Relation between the slope of the condition-specific linear regressions from (**I**) and the synaptic strength of the activated KC sets. (**K**) Relation between the slope of the condition-specific linear regressions from

*Figure 5 continued on next page*

*Figure 5 continued*

(**I**) and the number of recruited KCs. (**L**) Slopes of the condition-specific linear regressions from (**I**). Color codes of (**J–L**) as in (**I**). Statistical significance was tested in multiple comparisons with a parametric one-way ANOVA test (**D, E and H**) resulting in p=<0.0001 for all conditions, or in an unpaired parametric t-test (**F and G**) resulting in p=<0.0001 for all conditions. Numerical results are summarized in *Table 3*.

stochastic connectivity of projection neurons to KCs that relay odor information from the olfactory glomeruli to the MBONs (*Caron et al., 2013*; *Gruntman and Turner, 2013*; *Litwin-Kumar et al., 2017*; *Warth Pérez Arias et al., 2020*; *Zheng et al., 2022*). However, the role of MBONs within the circuit is largely fixed between animals and many MBONs can be classified as either approach or avoidance neurons in specific behavioral paradigms (*Aso et al., 2014a*; *Aso et al., 2014b*; *Litwin-Kumar et al., 2017*; *Li et al., 2020*; *Takemura et al., 2017*; *Scheffer et al., 2020*). Furthermore, individual KCs are not biased in their MBON connectivity but innervate both kinds of output modules, for both approach and avoidance. Learning and memory, the modulation of odor response behavior through pairing of an individual odor with either positive or negative valence, is incorporated *via* the local activity of valence-encoding DANs that can either depress or potentiate KC>MBON synaptic connections (*Aso and Rubin, 2016*; *Berry et al., 2018*; *Handler et al., 2019*; *Yamagata et al., 2015*; *Cohn et al., 2015*; *Burke et al., 2012*; *Hige et al., 2015a*; *Owald et al., 2015a*; *Jacob and Waddell, 2020*; *Pai et al., 2013*; *Plaçais et al., 2013*; *Séjourné et al., 2011*; *Bouzaiane et al., 2015*). As KC odor specificity and connectivity differ significantly between flies with respect to the number and position of KC>MBON synapses, this circuit module must be based on architectural features supporting robust formation of multiple memories regardless of specific individual KC connectivity.

Prior computational work addressing properties of central nervous system neurons in *Drosophila* relied on synthetic (randomly generated) data (*Gouwens and Wilson, 2009*) or partial neuronal reconstructions (*Scheffer et al., 2020*; *Tobin et al., 2017*). We go beyond these previous studies by combining precise structural data from the electron-microscopy based synaptic connectome (*Takemura et al., 2017*) with functional (electrophysiological) data. The structural data consists of the neuroanatomical structure of MBON-α3, including the 12,770 synaptic inputs of all its 948 innervating KCs. While this EM reconstruction may contain some mistakes in synaptic connectivity as e.g. up to 10% of synaptic sites remained unassigned within the dataset (*Takemura et al., 2017*), it currently represents the best possible template for an in silico reconstruction. The neuron's functional properties were determined from ex vivo patch clamp recordings. Near-perfect agreement between

**Table 3.** Summary of data from the plasticity simulations of *Figure 5*.
Simulations vary the number of active KCs and/or the strength of the KC-MBON synapses by scaling the maximal conductance parameter of the synaptic alpha-conductance function. Results include the mean number of activated synaptic sites as well as the mean amplitude of the somatic voltage depolarization across the 1000 trial repetitions of each simulation. Ranges are reported as parentheticals. The slope of the linear regression describing the relation between the number of active synapses and somatic voltage amplitude is reported (*Figure 5I–L*).

| Simulation parameters | | Results | | |
|---|---|---|---|---|
| # KCs | Synaptic strength | Mean # Activated synapses | Mean somatic depolarization amplitude [$mV$] | Slope |
| 63 | 125% | 847.88 (747 - 939) | 21.55 (19.66–23.21) | 0.0184 |
| 50 | 125% | 672.55 (573 - 778) | 18.14 (15.99–20.29) | 0.0209 |
| 38 | 125% | 511.77 (425 - 615) | 14.60 (12.48–16.96) | 0.0235 |
| 63 | 100% | 847.08 (725 - 977) | 18.26 (16.15–20.34) | 0.0166 |
| 50 | 100% | 673.76 (574 - 805) | 15.24 (13.34–17.58) | 0.0185 |
| 38 | 100% | 512.48 (429 - 599) | 12.14 (10.43–13.85) | 0.0203 |
| 63 | 75% | 849.05 (749 - 965) | 14.56 (13.11–16.17) | 0.0141 |
| 50 | 75% | 672.86 (570 - 767) | 11.98 (10.37–13.41) | 0.0153 |
| 38 | 75% | 511.97 (420 - 606) | 9.44 (7.91–10.96) | 0.0164 |

experimentally observed and simulated voltage traces recorded in the soma shows that linear cable theory is an excellent model for information integration in this system.

Together, we obtain a realistic in silico model of a central computational module of memory-modulated animal behavior, that is a mushroom body output neuron.

Our data show that the dendritic tree of the MBON is electrotonically compact, despite the complex architecture that includes a high degree of branching. This data is in agreement with a prior electrophysiological characterization of an MBON in locust (*Cassenaer and Laurent, 2007*), and a similar feature has been reported for neurons in the stomatogastric ganglion of crayfish. Here the electrotonic compactness supports linear integration of synaptic inputs across extensive arborizations and likely serves to functionally compensate for inter-individual variability (*Otopalik et al., 2017*; *Otopalik et al., 2019*). The location of an individual synaptic input within the dendritic tree has therefore only a minor effect on the amplitude of the neuron's output, despite large variations of local dendritic potentials. This effect was particularly striking when we restricted the analysis to a population of KCs with identical numbers of synapses that all elicited highly stereotypical responses. The compactification of the neuron is likely related to the architectural structure of its dendritic tree. Together with the relatively small size of many central neurons in *Drosophila*, this indicates that in contrast to large vertebrate neurons, local active amplification or other compensatory mechanisms (*Rumsey and Abbott, 2006*; *Sterratt et al., 2012*; *Froemke et al., 2005*; *Gidon and Segev, 2009*; *Schiller et al., 2000*; *Polsky et al., 2004*) may not be necessary to support input normalization in the dendritic tree. In contrast, for axons it has been recently reported that voltage-gated Na+ channels are localized in putative spike initiation zones in a subset of central neurons of *Drosophila* (*Ravenscroft et al., 2020*). In case in vivo physiological data quantitatively describing local active currents become available, they can be incorporated into our model to further increase the agreement between model and biological system.

Encoding of odor information and incorporation of memory traces is not performed by individual KCs but by ensembles of KCs and MBONs. Calcium imaging in vivo demonstrated that individual odors evoke activity reliably in approximately 3–9% of KCs (*Campbell et al., 2013*; *Honegger et al., 2011*). Our simulations of 1000 independent trials with random sets of 50 KCs, each representing one distinct odor that activates approximately 5% of the KCs innervating the target MBON, demonstrate that such activation patterns robustly elicit MBON activity in agreement with in vivo observations of odor-induced activity that elicited robust increases in action potential frequency in MBONs (*Hige et al., 2015a*; *Warth Pérez Arias et al., 2020*). The low variability of depolarizations observed in these simulations indicates that information coding by such activation patterns is highly robust. As a consequence, labeled line representations of odor identity are likely not necessary at the level of KCs since relaying information *via* any set of ≈50 KCs is of approximately equal efficiency. Such a model is supported by a recent computational study demonstrating that variability in parameters controlling neuronal excitability of individual KCs negatively affects associative memory performance. The authors provide evidence that compensatory variation mechanisms exist that ensure similar activity levels between all odor-encoding KC sets to maintain efficient memory performance (*Abdelrahman et al., 2021*).

Optical recordings of in vivo activity of MBONs revealed selective reductions in MBON activity in response to aversive odor training (*Jacob and Waddell, 2020*; *Pai et al., 2013*; *Owald and Waddell, 2015*; *Perisse et al., 2016*; *Séjourné et al., 2011*) or to optogenetic activation of selective DANs (*Hige et al., 2015a*). More generally, both depression and potentiation of MBON activity have been previously observed in different MBON modules in vivo (*Owald and Waddell, 2015*; *Hige et al., 2015a*; *Perisse et al., 2016*; *Plaçais et al., 2013*; *Pai et al., 2013*; *Séjourné et al., 2011*; *Jacob and Waddell, 2020*). In addition, recent studies have observed changes in KC stimulus representations after conditioning (*Delestro et al., 2020*; *Shyu et al., 2019*) that may be due to learning-dependent modulations of synaptic PN input to KCs (*Baltruschat et al., 2021*). Our computational model allowed us to implement and compare these two mechanisms changing MBON output: One is a change in the strength of the KC>MBON synapses (over a range of ±25%); the other is a change in the number of activated KCs (over the same range). Interestingly, we found almost linear relationships between the number of active KCs and the resulting depolarizations, and the same between the strength of synapses and MBON depolarization. Decreasing or increasing either of these variables by 25% significantly altered the level of MBON activation with only minor differences in the extent to which these

modifications contributed to MBON depolarization. The two different mechanisms of altering MBON output are potentially utilized for the establishment of different types of memory with fast and local alterations of synaptic transmission likely essential for short-term memory while structural changes may ensure maintenance of memory over long periods of time. In addition, differential modes of plasticity may be required at potentially more static parts of the MB circuitry like the food-related part that is not entirely based on stochastic connectivity (*Zheng et al., 2022*).

Our simulation data thus shows that the KC>MBON architecture represents a biophysical module that is well-suited to simultaneously process changes based on either synaptic and/or network modulation. Together with the electrotonic nature of the MBONs, the interplay between KCs and MBONs thus ensures reliable information processing and memory storage despite the stochastic connectivity of the memory circuitry. While our study focuses on the detailed activity patterns within a single neuron, the availability of large parts of the fly connectome at the synaptic level, in combination with realistic models for synaptic dynamics, should make it possible to extend this work to circuit models to gain a network understanding of the computational basis of decision making.

## Materials and methods

### Fly stocks

Flies were reared at 25°C and 65% humidity on standard fly food. The following stocks were used in this study: 10XUAS-IVS-mCD8::GFP (*Pfeiffer et al., 2010*) (BDSC 32186), MB082C-Gal4 (*Aso et al., 2014b*) (BDSC 68286), 2xUAS-EGFP (BDSC 23867) (all from the Bloomington *Drosophila* Stock Center, Indiana).

### Immunohistochemistry

Male flies expressing GFP in MBON-α3 were fixed for 3.5 hr at 4°C in 4% PFA containing PBST (0.2% Triton-X100). Flies were then washed for at least 3x30 min at RT in PBST before dissection. Dissected brains were then labelled with primary antibodies rabbit anti-GFP (A6455, Life technologies, Carlsbad, USA 1:2000) and mouse anti-Brp (nc82, Developmental Studies Hybridoma Bank, Iowa, USA, 1:200) for two nights at 4°C. After incubation, brains were washed at least 6x30 min in PBST. Secondary antibodies (Alexa488 (goatαrabbit) and Alexa568 (goatαmouse) coupled antibodies, Life technologies, Carlsbad, USA, 1:1000) were applied for two nights at 4°C. After a repeated washing period, brains were submerged in Vectashield (Vector Laboratories, Burlingame, USA), mounted onto microscope slides, and stored at 4°C.

Images were acquired using a LSM 710 confocal scanning microscope with a 25 x Plan-NEOFLUAR, NA 0.8 Korr DIC, oil objective (Carl Zeiss GmbH, Jena, Germany) and a Leica STELLARIS 8 confocal microscope with a 20 x HC PL APO NA 0.75 multi-immersion objective (Leica Microsystems, Wetzlar, Germany). Raw images were projected with Fiji (*Schindelin et al., 2012*) and cropped in Photoshop (Adobe, San José, USA). Uniform adjustments of brightness and contrast were performed.

### Electrophysiology

All patch clamp recordings were performed at room temperature similar to prior descriptions (*Hige et al., 2015a*; *Hige et al., 2015b*). Flies were flipped 1d after hatching to obtain 2- to 3-day-old flies. Due to the ventral location of the MBON-α3 somata within the brain, we performed the recordings ex vivo. For preparation, flies expressing GFP in MBON-α3 were briefly anesthetized on ice before removing the entire brain out of the head capsule. The preparation was performed in oxygenated (95% and 5%) high glucose ($260mM$) extracellular saline. Analogous to a prior study, the brains were incubated for $30s$ in $0.5mg/ml$ protease (from *Streptomyces griseus*, CAS# 90036-06-0, Sigma-Aldrich, St. Louis, USA) containing extracellular saline (*Wilson et al., 2004*). The brain was then transferred to standard extracellular saline (*Wilson et al., 2004*; in $mM$: 103 $NaCl$, 3 $KCl$, 5 TES, 10 trehalose, 10 glucose, 7 sucrose, 26 $NaHCO_3$, 1 $NaH_2PO_4$, 1.5 $CaCl_2$, 4 $MgCl_2$, pH 7.3, $280 - 290mOsmol/kg$ adjusted with sucrose). For physiological recordings, the brain was transferred to a glass bottom chamber with continuously perfused ($2ml/min$) oxygenated standard extracellular saline and held in place by a custom made platinum frame.

We used glass capillaries (GB150 (F-)–10 P, Science Products, Hofheim, Germany) and a horizontal puller (Flaming Brown Micropipette Puller P-1000, Sutter instruments, Novato, USA) to obtain

pipettes with a resistance of $7 - 10M\Omega$. The puller was equipped with a 2.5 mm box filament and patch pipettes were pulled with a program consisting of 2 lines (Line 1/Line 2: Heat 472/463 with a ramp value of 468, Pull 0/41, Velocity 52/70, Delay 1/10, Pressure 350). For recordings, patch pipettes were filled with internal solution (**Mauss et al., 2014**) (in $mM$: 140 K-aspartate, 10 HEPES, 4 Mg-ATP, 0.5 Na-GTP, 1 EGTA, 1 $KCl$, pH 7.29 adjusted with $KOH$, $265mOsmol/kg$). Whole-cell recordings were made using the EPC10 amplifier (HEKA, Reutlingen, Germany) and the PatchMaster software (HEKA, Reutlingen, Germany). Signals were low-pass filtered at $3kHz$, and digitized at $10kHz$ via a digital-to-analog converter. The liquid junction potential of $13mV$ was corrected online. Patch pipettes were guided under visual control with an upright microscope (BX51WI; Olympus, Tokio, Japan) equipped with a 60x water immersion objective (LUMPlanFl/IR; Olympus, Tokio, Japan) and a CCD camera (Orca 05 G, Hamamatsu, Japan). GFP signal of target cells was visualized through fluorescence excitation (100 W fluorescence lamp, Carl Zeiss GmbH, Jena, Germany), and a dual-band emission filter set (512/630 HC Dualband Filter GFP/DsRed, AHF Analysentechnik, Germany). We used GFP expression in soma and neurite to identify MBON-α3 neurons. During recordings, the fluorescence excitation was shut off to minimize phototoxic effects. The complete setup was mounted on an air-damped table while being shielded by a Faraday cage. Series resistance was maintained below $90M\Omega$ and compensated for up to 35% through the amplifier's compensation circuitry. In current-clamp mode, cells were held at a baseline of $-60mV$ to measure the relevant parameters for the model. Signals were recorded with a sample rate of $20kHz$ or $50kHz$ and low-pass filtered at $5kHz$.

To determine passive electrical membrane properties and qualitative patch characteristics, a series of protocols were performed:

### Protocol 1

In voltage-clamp mode, a voltage-step ($100ms$, $5mV$) is applied without compensating the series resistance. The corresponding current trace was then used to determine $R_{series}$ and $R_{input}$ using Ohms law and $R_m$ was determined as ($R_{series} - R_{input}$) (**Numberger and Draguhn, 1996**) in Igor Pro (WaveMetrics, Portland, USA).

### Protocol 2

In current-clamp mode, a step-protocol was performed and recorded at $50kHz$. Each sweep starts with $0pA$ for $10ms$, which is followed by a current injection ($-26pA$ to $32pA$, with $2pA$ increments) and a step duration of $400ms$. The resulting changes in membrane potential and induced action potentials were then analyzed and plotted with Matlab (MathWorks, Natick, USA). The maximum depolarization was calculated as a mean of the last 70 sample points of the stimulation sweep in each trace. The resting membrane potential was calculated as the mean of the first 500 sample points before the stimulation in each trace. The absolute action potential was measured as the maximum of each trace. The relative action potential amplitude resulted from subtracting the baseline from the absolute value. The baseline resulted from the mean of a 400 sample points window, starting 700 sample points upstream of the maxima in each trace. The resulting trace of the $-10pA$ injection was used to calculate the membrane time constant $\tau_m$ of each cell.

### Protocol 3

In current-clamp mode, no current was injected throughout a $60s$ recording at $50kHz$. The resting membrane potential was determined as the mean of the baseline in the recording with Igor Pro and spontaneous activity was plotted with Matlab.

### Protocol 4

In current-clamp mode, a short current pulse ($-10pA$, $200ms$) was injected and recorded at $20kHz$. We performed 35 iterations per cell. The resulting data of three example cells was used for fitting the membrane kinetics of the model as described previously (**Gouwens and Wilson, 2009**). Averaging, normalisation (($V_m$-$V_{m0}$)/$V_{m0}$) and plotting was performed with Matlab. This data was used to perform the model fitting. The baseline ($V_{m0}$) resulted from the average of the sample points before stimulation.

Cells displaying a resting membrane potential higher than $-45mV$ and/or where the series resistance was too high (> $90M\Omega$) were excluded from the analysis. Only cells that were firing action potentials were included in the analysis. A list of the five included cells using mCD-GFP as a marker and of the four cells using cytoplasmic GFP as a marker and the corresponding calculated values are provided in *Table 1* and *Table 1—source data 1*.

## Computational model

Morphology data for the *Drosophila* MBON-α3-A was originally characterized using scanning electron microscopy (*Takemura et al., 2017*) and for the present study obtained from the 'mushroombody' dataset hosted in the database neuPrint (https://neuprint-examples.janelia.org/; cell ID 54977). These data describe the structure of its dendritic arborization (*Figure 2B*, and *Figure 2—figure supplement 1*), with coordinates specified at 8 nm pixel resolution and converted to $\mu m$ for model implementation. The reconstruction is limited to the portion of the neuron in the mushroom body, not including its soma and axon. The geometry of the axon was determined from the corresponding MBON14(α3) R (*Figure 2A*), obtained from neuPrint's 'hemibrain:v1.1' dataset (https://neuprint.janelia.org/; cell ID 300972942). The axon and synaptic terminal of this neuron were approximated as five separate sections, distinguished by the major branch points identified in the electron microscopic reconstruction (*Figure 2—figure supplement 1*). Each section was characterized by its total length and average diameter and divided into segments to maintain the average section length of that region in the original reconstruction. They were then appended to the MBON-α3-A proximal neurite as individual linear segments (*Figure 2—figure supplement 1*). The dimensions of the soma were characterized by confocal microscopy of MBON-α3-A (*Figure 1A*). Synaptic locations of the KC innervation of MBON-α3-A were also obtained from neuPrint's 'mushroombody' dataset. MBON-α3-A is innervated by 948 distinct KCs, forming a total of 12,770 synapses with an average of 13.47 synapses per KC. This data set potentially misses up to 10% of synaptic connections and includes errors associated with electron microscopy reconstructions (*Takemura et al., 2017*).

Electrophysiological properties of the model neuron (*Table 2*) were determined by fitting the model response to the recorded data, an approach consistent with prior work modeling *Drosophila* central neurons (*Gouwens and Wilson, 2009*). Somatic voltage excursions in response to $200ms$ current injections of $-10pA$ were recorded ex vivo (*Figure 1G*) and compared with the simulated membrane potential (*Figure 2D*). This current injection was replicated in silico and the membrane potential change at the soma was fit to our recorded data by varying the cell's cytoplasm resistivity, specific membrane capacitance, and passive membrane conductance. The fitting was performed using NEURON's principle axis optimization algorithm, where parameters were tuned until the computed voltage excursion at the soma matched our electrophysiological recordings. The model optimization searched parameter space to minimize an error function defined as the squared differences between the averaged experimental responses to current pulses (*Figure 1G*) and the computed model responses (*Figure 2D–F*). The fitting was then verified by normalizing both the model and experimental voltage excursions. The model was first allowed to equilibrate for $30ms$ simulated time, after which the $200ms$ current pulse was initiated. The region of data used for the fitting ranged from $32.025ms$ to $228.03ms$, excluding approximately the first and last $2ms$ of current injection. The domain of the parameter space was restricted to physiologically plausible values: $0.5\mu F/cm^2$ to $1.5\mu F/cm^2$ for specific membrane capacitance, $1E-7 S/cm^2$ to $1E-4 S/cm^2$ for passive membrane conductance, and $30\Omega cm$ to $400\Omega cm$ for cytoplasm resistivity (*Table 2*). The final values for each parameter are well within these margins. Resting membrane potential and the leak current reversal potential were set at $-55.64mV$ based on the initial conditions of the recorded data to which the model was fit.

Cholinergic KC to MBON synapses were modeled as localized conductance changes described by alpha functions (*Table 2*). KC innervation of MBON-α3-A was determined to be cholinergic (*Barnstedt et al., 2016*). Currents recorded in *Drosophila* KCs were demonstrated to have a reversal potential of $8.9mV$ and a rise time to a maximal conductance of $0.44ms$ (*Su and O'Dowd, 2003*). The maximal conductance for the synapses was set at $1.5627*10^{-5}\mu S$, the value determined to achieve the target MBON depolarization from monosynaptic KC innervation (*Hige et al., 2015b*).

All simulations were performed using the NEURON 7.7.2 simulation environment (*Hines and Carnevale, 1997*) running through a Python 3.8.5 interface in Ubuntu 20.04.1. The morphology data divided the MBON-α3-A dendritic tree into 4336 sections. We added 6 additional sections describing

**Table 4.** Morphological parameters of the model.

| Parameter | Value |
| --- | --- |
| Number of sections | 4342 |
| Average section length | $1.24 \mu m$ |
| Total length | $5377.39 \mu m$ |
| Average diameter | $0.29 \mu m$ |
| Average segment surface area | $1.26 \mu m^2$ |
| Total surface area | $6168.32 \mu m^2$ |
| Soma diameter | $6.45 \mu m$ |

the axon, synaptic terminals, and soma to a total of 4342 sections with an average of $1.24 \mu m$ in length and $0.29 \mu m$ in diameter (*Table 4*). Passive leak channels were inserted along the membrane of the dendrite sections as well as throughout the axon and soma. Each dendritic section was automatically assigned an index number by NEURON and modeled as a piece of cylindrical cable, with connectivity specified by the neuron's morphology. The coordinate location of each KC synapse was mapped to the nearest section of the MBON dendrite, and the conductance-based synapse model was implemented in the center of that section. Simulations were performed in which each synapse-containing section was individually activated, followed by additional simulations activating groups of synapses corresponding to single or multiple active KCs. A list of all performed simulations related to *Figure 5* is provided in *Table 3*. Membrane potential data was analyzed and plotted with MATLAB R2020b (MathWorks, Natick, USA) and Prism 9 (GraphPad, San Diego, USA).

All code and data files necessary to replicate the simulations are available at: https://doi.org/10.7281/T1/HRK27V.

## Statistical analysis

Statistical analysis and visualization of data was performed in Prism 9 (GraphPad, San Diego, USA) and Matlab (MathWorks, Natick, USA). Before a statistical comparison was performed, individual groups were tested in Prism for normality and lognormality with an Anderson-Darling test. Statistical significance was tested with an unpaired parametric t-test or in case of multiple comparisons, with an ordinary corrected parametric one-way ANOVA test. Further information is provided in figure legends.

## Acknowledgements

We thank Glenn Turner and Toshihide Hige for insightful comments on the patch clamp analysis, Hans-Peter Schneider for technical help and Oliver Barnstedt for comments on the manuscript. Funding: This work was supported by NIH R01DC020123, NSF 1835202, NIH R01DA040990, NIH R01EY027544 (all EN), NIH Medical Scientist Training Program Training Grant T32GM136651 (OH), and a BMBF (FKZ 01GQ2105) and a DFG grant (INST 248/293–1) (all JP).

## Additional information

### Funding

| Funder | Grant reference number | Author |
| --- | --- | --- |
| National Institutes of Health | R01DC020123 | Ernst Niebur |
| National Institutes of Health | R01DA040990 | Ernst Niebur |

| Funder | Grant reference number | Author |
| --- | --- | --- |
| National Institutes of Health | R01EY027544 | Ernst Niebur |
| National Institutes of Health | Medical Scientist Training Program Training Grant T32GM136651 | Omar A Hafez |
| National Science Foundation | 1835202 | Ernst Niebur |
| Bundesministerium für Bildung und Forschung | FKZ 01GQ2105 | Jan Pielage |
| Deutsche Forschungsgemeinschaft | INST 248/293-1 | Jan Pielage |

The funders had no role in study design, data collection and interpretation, or the decision to submit the work for publication.

### Author contributions

Omar A Hafez, Conceptualization, Data curation, Software, Formal analysis, Validation, Visualization, Methodology, Writing – original draft, Writing – review and editing; Benjamin Escribano, Rouven L Ziegler, Conceptualization, Data curation, Formal analysis, Validation, Visualization, Methodology, Writing – original draft, Writing – review and editing; Jan J Hirtz, Formal analysis, Methodology; Ernst Niebur, Conceptualization, Data curation, Software, Formal analysis, Supervision, Funding acquisition, Validation, Visualization, Writing – original draft, Project administration, Writing – review and editing; Jan Pielage, Conceptualization, Data curation, Formal analysis, Supervision, Funding acquisition, Validation, Visualization, Methodology, Writing – original draft, Project administration, Writing – review and editing

### Author ORCIDs

Omar A Hafez ![ORCID] http://orcid.org/0000-0002-7846-9226
Benjamin Escribano ![ORCID] http://orcid.org/0000-0002-1432-5952
Rouven L Ziegler ![ORCID] http://orcid.org/0000-0002-3050-7692
Jan J Hirtz ![ORCID] http://orcid.org/0000-0002-4486-3057
Ernst Niebur ![ORCID] http://orcid.org/0000-0002-2815-9262
Jan Pielage ![ORCID] http://orcid.org/0000-0002-5115-5884

### Decision letter and Author response

Decision letter https://doi.org/10.7554/eLife.77578.sa1
Author response https://doi.org/10.7554/eLife.77578.sa2

## Additional files

### Supplementary files
• Transparent reporting form

### Data availability

All data generated or analysed in this study are included in the manuscript. All simulation files and the code and data files needed to replicate the simulations are available as a permanent and freely accessible data collection at the Johns Hopkins University Data Archive: https://doi.org/10.7281/T1/HRK27V. This includes the simulation code itself (python), the structural EM reconstruction of MBON-alpha3 (swc), the EM reconstruction of the related MBON used to model the axon and synaptic terminal structures (swc), the synapse locations as coordinate data (json), and the synapse locations by MBON section (json). Parameter values for model definition and individual simulations are specified within the code files and outlined in each figure legend where appropriate.

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
