## [Editor Report]

In light of the ongoing emergence of volume electron microscopy connectomics, detailed morphologies at the nanometre scale for many neurons are now available, ready for functional and computational analysis. Building on these foundational resources, this work delivers compelling evidence that synaptic inputs onto the dendritic arborisations of readout neurons (MBONs) of the learning and memory system of the adult *Drosophila melanogaster* contribute with equal weight to the depolarization of the neuron, independently of their location on the arbor, a phenomenon known as synaptic democracy. These important findings establish the validity of computational models based on passive dendritic propagation for simulating fly brain circuits and highlight the differences between the much larger mammalian neurons that present active propagation strategies as part of their approach to synaptic democracy.

---

## [Decision Letter]

**Decision letter after peer review:**

Thank you for submitting your article "The cellular architecture of memory modules in *Drosophila* supports stochastic input integration" for consideration by *eLife*. Your article has been reviewed by 3 peer reviewers, and the evaluation has been overseen by a Reviewing Editor and K VijayRaghavan as the Senior Editor. The following individual involved in review of your submission has agreed to reveal their identity: Albert Cardona (Reviewer #1).

Essential revisions:

Please follow carefully the many detailed comments by the reviewers. To emphasize comments with regard to the caveats of using partially EM-reconstructed neuronal morphologies, and in the statements regarding electrotonic compactness.

*Reviewer #1 (Recommendations for the authors):*

Hafez and collaborators describe the construction and analysis of a computational model of a mushroom body neuron. The anatomy derives from a combination of electron microscopy reconstructions of MBON-α3 and also from light microscopy. The physiological parameters derive from publications that measured them, in addition to the author's own electrophysiological recordings with patch-clamp.

There are two main findings. First, the dendritic arbor of MBON-α3 is electrotonically compact, meaning, individual connections from Kenyon cells will similarly elicit action potentials independently as to where, spatially, the synapses lay on the arbor. Second, in simulation, exploration of changes in the strength of Kenyon cell inputs illustrate two possible ways to alter the strength of the KC-MBON physiological connection, showing that either could account for the observed synaptic depression in the establishment of associative memories. The properties of each approach differ.

Overall, the manuscript clearly describes the journey from connectomics and electrophysiology to computational modeling and exploration of the physiological properties of a circuit in simulation.

The discussion ought to be expanded to include the implications of two possible approaches to physiologically altering the KC-MBON synapse and the consequence of their combination in expanding the space of alterations induced by associative memory paradigms.

In general, the results are clear, but some details remain underdetermined and I have listed them below in the detailed comments. The introduction and discussion present some inaccuracies that can be swiftly addressed by the authors

Detailed comments:

Line 45: Language: "potential rewarding": potentially.

Line 49: Language: "cellular and circuit architecture contributes": architectures contribute.

Line 72: instead of 5%, the number of KCs active at any one time seems to be 6% as per Turner et al. 2008 and Campbell et al. 2013. What is the robustness of the analysis to this small change? Did you explore a range of possible single-digit percent KC activations?

Line 99: Aso & Rubin 2016 belongs with citations in line 95.

Line 122: "*Drosophila*" needs italics, throughout the manuscript.

The authors devise a computational model of an MBON using a neuronal arbor reconstructed from volume electron microscopy by Takemura et al. 2017. That paper details that only 93% of all synapses were connected to an arbor, and only 86% of the synapses had known pre- and postsynaptic arbors. For the MBON that was used for modeling, what was the fraction of terminal ends labeled as uncertain, and where these clustered or scattered across the arbor? Furthermore, the volume imaged with FIBSEM did not fully enclose the vertical lobe of the MB. Any estimate of what fraction of the chosen MBON's arbor is contained within the imaged volume? In other words, what analysis has been done here to ensure that the modeled arbor is representative of an MBON arbor in vivo, and what mitigating measures were taken to account for the potentially missing 14% or more of the arbor synapses and terminal dendrites?

The authors report using the 10XUAS-IVS-mCD8::GFP to label the MBON, so that they can then record electrophysiologically with patch-clamp. What is the effect of inserting so many mCD8 proteins (a large transmembrane protein) into the neuron's membrane on the voltage potential and action potential formation and transport? The 10XUAS is particularly strong. How does the morphology of the imaged neuron differ from that of the EM-reconstructed neuron, regarding calibers and amount of cable? For this purpose, a cytosolic GFP targeting the soma or nucleus and poorly diffusing into the arbor would have been far preferable, as the effect of inserting transmembrane proteins in neurons' membranes on resting potentials is well reported.

Line 155: average resting potential for the MBON is reported at -56.7 mV +/- 2.0 mV. In Hige et al. 2015a, cells were held at -70 or -60 mV. Nowhere does Hige et al. 2015a report on the resting potential.

Line 174: amplitude of action potentials was rather small, but in Hige et al. 2015a action potentials typically exceeded 200 pA. Is this what the authors mean by small? Just how small were the recorded action potentials?

Line 176: by small amplitudes and the explanation on the long neurite connecting the dendritic arbor with the soma, you mean that the signal is attenuated over long distances?

Line 179: how was the membrane capacitance calculated?

Table 1: 5 neurons were used. How do you know they are all MBON-α3? Has it been confirmed that the GAL4 line doesn't have stochastic expression among similar yet different sibling neurons of the same lineage? How many other MBONs innervate the tip of the α lobe and do any of them share neuroblast lineage with MBON-α3? The large differences in measured values listed in Table 1 could be explained by having measured similar yet different neurons. Did you run a battery of tests before and after the measurements to ensure the recorded neurons remained in good health throughout the measurement session? Such tests often consist in a ramp of current injections and the recording of the neuron's responses, which are then compared between before and after the experimental measurements of membrane properties (like the current step protocol of Figure 1F).

Line 184: why only 3 cells? In *Drosophila*, recording from e.g. 10 cells, all homologous cells across 10 individuals, gives e.g., 8 responding with excitation to a sensory stimuli with some variation and 2 responding with inhibition. There is a lot of variability in the responses. Recording from only 3 cells seems risky, statistically speaking. What justifies this low number?

Line 186: "To to further".

Line 199: when you say that the measurements are in "good agreement with prior recordings" of other neurons in *Drosophila*, what do you mean exactly? How similar, how far off, by what parameters?

Line 203: might as well mention that there were 948 reconstructed KCs synapting onto MBON-α3, so 5% is 50. Spare the reader remembering where the 50 was picked from. (If you correct this to 6%, would be 57 KCs). And you seem to not keep in mind that the KCs responding to a specific odor may be correlated in their synaptic connectivity strength onto MBON-α3. Data to this end may be included in Li et al. 2020 *eLife* where the whole mushroom body is reconstructed, including the olfactory projection neurons, so such correlations if any may be evident in that data set.

Line 210: a complete reconstruction of MBON-α3 now exists, from either the FAFB volume or the Hemibrain volume. In the methods you mention you used the Hemibrain data set for the axon.

Line 209: the 12,770 synaptic connections aren't "all", these are the ones reported from the anatomical reconstruction from volume electron microscopy. According to the source papers (Takemura et al. 2017) about 10% of all synapses are missing. An analysis of how these missing synapses impacts the structure of the arbor is absent from the paper.

In addition, sample preparation for electron microscopy with chemical fixation alters the fine anatomical details, including the length of terminal dendrites and the calibers of neurites throughout. See e.g., Korogod et al. 2015 *eLife* "Ultrastructural analysis of adult mouse neocortex comparing aldehyde perfusion with cryo fixation" and the follow up paper Tamada et al. 2020 *eLife* "Ultrastructural comparison of dendritic spine morphology preserved with cryo and chemical fixation". What measures were taken to correct or mitigate these artifactual differences with in vivo neurons?

Later, Figure 2F strongly supports the appropriateness of the model, yet, the above points merit discussion and even exploration: how much of the dendritic arbor can you miss and still get the same result? What does the response to current injection depend on, cable, number of synapses, synapse spatial location, cable calibers, tapering of cable? What cable truncations are tolerable? This is very important information towards future computational studies based on neuronal morphologies reconstructed from volume electron microscopy.

Figure 2 legend: what is the evidence that the "proximal neurite" in green in Figure 2B is the site of axon potential generation? Gouwens and Wilson 2009 pointed at a region anywhere between the root of the dendritic tree and half-way through the axon of the uniglomerular olfactory projection neuron they modeled.

Does the site of axon potential generation emerge from your model, or did you specify it in the model?

Why is the two-tailed non-parametric Spearman correlation the correct statistic to compare the modeled and the experimentally measured membrane potential in Figure 2F?

Figure 2 legend reads "see appendix" but there isn't any appendix to the manuscript?

Line 249: "the average number of synaptic contacts from a KC to this MBON is 13.47". This statement ought to be qualified: for the single MBON-α3 measured in Takemura et al. 2017, and with the caveat of ~10% of synapses potentially missing. You could just as easily apply a correction factor and say the average number is about 14.7 + 1.47 = 16.2. Would this change the outcome of your model?

Please don't use "PN" as an acronym for proximal neurite. First, *eLife* doesn't restrict the length of your test. Second, PN is an established acronym, universally across all neuroscience literature, for projection neuron. Plus, the "proximal neurite" (as per figure 2B) might as well be called the putative AIS (axon initial segment; pAIS for "putative") where the integration of inputs across the entire dendritic tree take place and the axon potential is initiated.

Figure 3H: in the measurement of "local dendritic section volume", did you correct for volume artifacts induced by using (in purpose!) an incorrect osmolarity of the buffers when fixing the tissue in the sample preparation protocol for electron microscopy? See Korogod et al. 2015 *eLife* and Tamada et al. 2020 *eLife*.

Line 291: "this value is in good agreement with in vivo data for MBONs". Please could you specify what this agreement is, how close, some details.

Line 293: in line with analyzing all KCs with exactly 13 synaptic inputs onto MBON-α3, what's the result of analyzing the voltage excursion from drawing random subsets of 13 synapses? (or 16 as per the correction, see above). Are the natural groups of 13 synapses different in their effect on the neuron's voltage than artificial groupings?

Line 299: inaccurate statement: "Given that ≈ 5% of the 984 KCs innervating MBON-α3 are typically activated by an odor". Instead, what is known from the literature is that, given the presence of the GABAergic neuron APL in the mushroom body which acts as the inhibitory unit of a winner-take-all configuration, only 6% (not 5%) of KCs simultaneously respond to any one odor. Plus when the APL neuron is inhibited, a huge double-digit percent of KCs are active in response to an odor.

Line 311: there is now far better evidence of stochastic odor encoding by KCs than Caron et al. 2013. See Zheng e t al. 2020 bioRxiv "Structured sampling of olfactory input by the fly mushroom body", and Li et al. 2020 *eLife*, and also, for larvae, Eichler et al. 2017 Nature.

Line 319: see also Baltruschat L, Prisco L, Ranft P, Lauritzen JS, Fiala A, Bock DD, Tavosanis G. Circuit reorganization in the *Drosophila* mushroom body calyx accompanies memory consolidation. Cell reports. 2021 Mar 16;34(11):108871.

Line 337: the differences in the somatic amplitudes may be significant statistically, but are they meaningful? In other words, the effect size looks like near zero. The real, and important difference, is in Line 345 where it is stated that "we observed some differences in the slope of the responses between the different tuning modalities (Figure 5J,K,L)."

Line 350: Scheffer et al. 2020 is not an appropriate citation for the statement "Te ability of an animal to adapt its behavior to a large spectrum of sensory information requires specialized neuronal circuit motifs". Rather, a textbook such as Kandel et al. Principles of Neural Science, or no reference at all, would be appropriate. You could also delete the sentence without loss.

Line 353: *Drosophila* must go in italics, it's a species name. Multiple occurrences throughout.

Line 356: through both short-term and long-term memory. A good example is Aso & Rubin 2016 *eLife*.

Line 359: note the olfactory system of the fly has a sort of "fovea", Zheng et al. 2020 bioRxiv.

Line 362: this sentence needs work, I am not sure what it means: "Individual flies display idiosyncratic, apparently random connectivity patterns that transmit information of specific odors to the output circuit of the MB."

Line 366: MBONs can only each be classified as approach or avoidance within specific behavioral paradigms. In different contexts, including different physiological states (e.g., hunger, satiation, others), the classification changes.

Line 381: prior work includes Tobin et al. 2017 *eLife*, where EM-reconstructed dendrites of olfactory projection neurons were modeled to understand the impact of dendritic arbor size on neuronal function.

Line 386: again, these numbers aren't precise. There's about 10% of missing synapses to consider, and potentially additional Kenyon cells. And some of the KCs, particularly those with low number of synaptic connections to MBON-α3, may have been connected in error.

Line 397: Strongest finding of this work: "The location of an individual synaptic input within the dendritic tree has therefore only a minor effect on the amplitude of the neuron's output, despite large variations of local dendritic potential." Would be best to surface it more.

Line 402: a comparison would be appropriate with neurons from the crayfish stomatogastric ganglion (STG) as described by Eve Marder lab's, with published findings such as neurons being electrotonically compact despite their large size in mature adult animals. For example, Otopalik et al. 2017 *eLife* "When complex neuronal structures may not matter", where the authors "quantify animal-to-animal variability in cable lengths (CV = 0.4) and branching patterns in the Gastric Mill (GM) neuron". And also Otopalik et al. 2019 *eLife* "Neuronal morphologies built for reliable physiology in a rhythmic motor circuit".

Line 406: you forgot to cite Jackie Schiller's work on cortical pyramidal cells and their tuft dendrites, some of which predates all the cited work in this statement.

Line 412: your model, as per Figure 2F, rather very closely matches the observed electrophysiological responses of the neuron under study. In what further ways could you model more closely match experimental observations? This would be very instructive to the reader.

Line 415: by ensembles of both KCS and MBONs, not just KCs, if you are including the memory part and not only the input representation part.

Line 416: for most of the paper, you quoted these papers to justify the 5% of KCs being simultaneously active in response to an odor. Now the range is shown as 3 to 9%. This discrepancy ought to be reconciled.

Line 422: again, nowhere in the manuscript so far did you detail in what way your simulation and experimental findings match those of prior experimental reports regarding neuron response and physiological properties.

Line 425: this statement is inaccurate. An individual KC does not encode for a single odor. That would almost never be the case even if a KC was single-claw, as in, exclusively integrated inputs from a single projection neuron: individual projection neurons rarely encode an odor; it's the population of projection neurons that does. Similarly, ensembles of co-active KCs together represent an odor, and do so more narrowly and accurately than the population of projection neurons that excited them.

Line 452: left unresolved remains the question of why would both mechanisms exist, as in, is the combination of altering the KC-MBON synapse and altering the PN-KC synapse better in some dimension than altering either alone? Does this relate perhaps to a possible dynamic nature of the olfactory "fovea" proposed by Zheng et al. 2020 bioRxiv as presumably static?

Line 466: what is standard fly food? Please define it. Choice of food affects very much behavioral assays, for example.

Line 476: no antigen saturation steps in the immunohistochemistry protocol? Please revise.

Line 505: what were the settings of the puller? These would be necessary to reproduce your glass capillaries. Did you grind the tips, and if so, how?

Line 534: how were the R_series, R_input and R_m determined with Igor Pro?

Line 561: why was a threshold of 600 MΩ used to exclude cells? How many cells were recorded in total, and how many were excluded?

Line 584: once again the data is the best human effort in proofreading a semiautomatic segmentation. It is not the absolute truth. Also, each individual fly is somewhat different, so this is merely one fairly complete yet partial reconstruction, and not assured to be error free, particularly of errors of omission, of a single MBON from a single individual. Would be appropriate to remark this clearly.

Did you provide the NeuroML or similar files necessary to run the model in the NEURON simulation software? These should be appended to the manuscript as supplemental data.

Line 596: if "parameters were tuned until the computed voltage excursion at the soma matched our electrophysiological recordings", what is the rationale for comparing the voltage potential of the simulation with those of the experimental observations?

Line 607: where do these ranges of physiologically plausible values come from? Citation, or measurements done in the lab and therefore a figure is needed to show them? Are these ranges from the literature listed in Table 2? Likely yes, would be appropriate to cite them here too or at least explicitly point to Table 2.

Line 616: innervation of MBONs is cholinergic because KCs are cholinergic, but that's not from Takemura et al. 2017, instead from Barnsted et al. 2016 Neuron.

Line 622: would be appropriate to include the python scripts used to configure and run the NEURON simulation as supplemental material. Likewise for the matlab scripts used to load the analyzed data and plot it. The raw data ought to be included as CSV files or similar.

*Reviewer #2 (Recommendations for the authors):*

"The cellular architecture of memory modules in *Drosophila* supports stochastic input integration" is a classical biophysical compartmental modelling study. It takes advantage of some simple current injection protocols in a massively complex mushroom body neuron called MBON–a3 and compartmental models that simulate the electrophysiological behaviour given a detailed description of the anatomical extent of its neurites.

This work is interesting in a number of ways:

– The input structure information comes from EM data (Kenyon cells) although this is not discussed much in the paper.

– The paper predicts a potentially novel normalization of the throughput of KC inputs at the level of the proximal dendrite and soma.

– It claims a new computational principle in dendrites, this didn't become very clear to me.

Problems I see:

– The current injections did not last long enough to reach steady state (e.g. Figure 1FG), and the model current injection traces have two time constants but the data only one (Figure 2DF). This does not make me very confident in the results and conclusions.

– The time constant in Table 1 is much shorter than in Figure 1FG?

– Related to this, the capacitance values are very low maybe this can be explained by the model's wrong assumption of tau?

– That latter in turn could be because of either space clamp issues in this hugely complex cell or bad model predictions due to incomplete reconstructions, bad match between morphology and electrophysiology (both are from different datasets?), or unknown ion channels that produce non–linear behaviour during the current injections.

– The PRAXIS method in NEURON seems too ad hoc. Passive properties of a neuron should probably rather be explored in parameter scans.

Questions I have:

– Computational aspects were previously addressed by e.g. Larry Abbott and Gilles Laurent (sparse coding), how do the findings here distinguish themselves from this work.

– What is valence information?

– It seems that Martin Nawrot's work would be relevant to this work.

– Compactification and democratization could be related to other work like Otopalik et al. 2017 *eLife* but also passive normalization. The equal efficiency in line 427 reminds me of dendritic/synaptic democracy and dendritic constancy.

– The morphology does not obviously seem compact, how unusual would it be that such a complex dendrite is so compact?

– What were the advantages of using the EM circuit?

– Isn't Figure 4E rather trivial if the cell is compact?

Overall, I am worried that the passive modelling study of the MBON–a3 does not provide enough evidence to explain the electrophysiological behaviour of the cell and to make accurate predictions of the cell's responses to a variety of stochastic KC inputs.

*Reviewer #3 (Recommendations for the authors):*

This manuscript presents an analysis of the cellular integration properties of a specific mushroom body output neuron, MBON-α3, using a combination of patch clamp recordings and data from electron microscopy. The study demonstrates that the neuron is electrotonically compact permitting linear integration of synaptic input from Kenyon cells that represent odor identity.

Strengths of the manuscript:

1) The study integrates morphological data about MBON-α3 along with parameters derived from electrophysiological measurements to build a detailed model.

2) The modeling provides support for existing models of how olfactory memory is related to integration at the MBON.

Weaknesses of the manuscript:

1) The study does not provide experimental validation of the results of the computational model.

2) The conclusion of the modeling analysis is that the neuron integrates synaptic inputs almost completely linearly. All the subsequent analyses are straightforward consequences of this result.

3) The manuscript does not provide much explanation or intuition as to why this linear conclusion holds.

In general, there is a clear takeaway here, which is that the dendritic tree of MBON-α3 in the lobes is highly electrotonically compact. The authors did not provide much explanation as to why this is, and the paper would benefit from a clearer conclusion. Furthermore, I found the results of Figures 4 and 5 rather straightforward given this previous observation. I am sceptical about whether the tiny variations in, e.g. Figures 3I and 5F-H, are meaningful biologically.

1) My biggest question is about the claim of extreme electrotonic compactness of this neuron. Figure 3D,E suggests that the voltage change at the proximal neurite and at the soma varies by only about 1% depending on stimulation location. Since this is supported only by simulation, it is worth asking how robust this conclusion is.

a) Given that the variability in 3I is so small in magnitude, any dependence would be swamped by other sources of heterogeneity, so the statement that this correlates with distance (line 278) is likely irrelevant.

b) Can the authors provide a confidence interval for the fit of biophysical parameters to their recordings?

c) On lines 271-274, the authors state, "This architecture with the smallest dendritic sections at the most distant sites may contribute to the compactness of the dendritic tree, ensuring that even the most distant synaptic inputs result in somatic voltage deflections comparable to the most proximal ones." Is a dependence of dendritic size on distance required for the results? It seems like the result is simply that there is no attenuation within the dendritic tree at all. In general, the authors don't actually provide an explanation for the compactness. In the discussion, it is stated that, "The compactification of the neuron is likely related to the architectural structure of its dendritic tree," which again is rather vague. The authors should strive to provide a clear explanation for this, since it is their key result.

d) Can the authors report what the electrotonic length for such a dendrite would be? How long before we expect to see a significant spread in 3E?

2) My other concern is that, once we assume this perfect integration, the subsequent analyses are all a straightforward consequence. In particular, Figures 4 and 5 just repeatedly convey that it doesn't matter which synapse is being activated, the effect on the somatic voltage is the same. I was particularly confused about the conclusions of 5F,G,H. The authors claim "small but significant differences" here, but practically speaking, I can't imagine any of the differences in these plots being meaningful.

3) Reference to the data used to constrain the model is confusing.

a) At various places in the manuscript references are made to in vivo recordings, but it appears that all of the recordings were done ex vivo.

b) On lines 389-392, the authors state: "Near-perfect agreement between experimentally observed and simulated voltage distributions in the dendritic tree shows that linear cable theory is an excellent model for information integration in this system." What recordings of the voltage distribution in the dendritic tree were performed?

4) It seems like the conclusions are different than those of Gouwens and Wilson (2009), who described their reconstructed PNs as electrotonically extensive. The authors should comment on what about MBONs and PNs is different.

---

## [Author Response]

Reviewer #1 (Recommendations for the authors):Hafez and collaborators describe the construction and analysis of a computational model of a mushroom body neuron. The anatomy derives from a combination of electron microscopy reconstructions of MBON-α3 and also from light microscopy. The physiological parameters derive from publications that measured them, in addition to the author's own electrophysiological recordings with patch-clamp.There are two main findings. First, the dendritic arbor of MBON-α3 is electrotonically compact, meaning, individual connections from Kenyon cells will similarly elicit action potentials independently as to where, spatially, the synapses lay on the arbor. Second, in simulation, exploration of changes in the strength of Kenyon cell inputs illustrate two possible ways to alter the strength of the KC-MBON physiological connection, showing that either could account for the observed synaptic depression in the establishment of associative memories. The properties of each approach differ.Overall, the manuscript clearly describes the journey from connectomics and electrophysiology to computational modeling and exploration of the physiological properties of a circuit in simulation.The discussion ought to be expanded to include the implications of two possible approaches to physiologically altering the KC-MBON synapse and the consequence of their combination in expanding the space of alterations induced by associative memory paradigms.

We now extended this discussion. However, as we do not provide any experimental evidence supporting either one of these hypothesis, we think that this point would be better addressed in a review format as we can only speculate at this point of time.

In general, the results are clear, but some details remain underdetermined and I have listed them below in the detailed comments. The introduction and discussion present some inaccuracies that can be swiftly addressed by the authorsDetailed comments:Line 45: Language: "potential rewarding": potentially.

We have changed the text accordingly.

Line 49: Language: "cellular and circuit architecture contributes": architectures contribute.

We have changed the text accordingly.

Line 72: instead of 5%, the number of KCs active at any one time seems to be 6% as per Turner et al. 2008 and Campbell et al. 2013. What is the robustness of the analysis to this small change? Did you explore a range of possible single-digit percent KC activations?

We now state the number as 3-9% of KCs and highlight that different studies observed reliable calcium signals in 5 or 6% of KC upon odor stimulation. Our own results (Siegenthaler et al. 2019) indicate that approximately 5% of KCs are activated by each odor and we used this number as an approximation for our simulations. It is correct, however, that the single-cell study by Turner et al. (2008) reported an average 6% of activated KCs for each odor. A later optogenetical study by the same group (Honegger et al., 2011) reported 5%; importantly for the present discussion, they however considered this result "extremely similar" to their earlier results (their page 11,777, left column). We take from their formulation that they consider the difference between 5% and 6% as very small It would then make sense to assume that small differences obtained in different experimental preparations, on the order of a percentage points, as a kind of noise, and to evaluate the robustness of results under such perturbations. However, in absolute terms this difference by one percentage point, between 5% and 6%, represents a fifth difference (i.e., 20%) in the synaptic input to the MBON. And, of course, in our results this difference can not be due to different experimental conditions since they are obtained in the same "experimental preparations," namely the same simulations. To answer directly the Reviewer’s question, we have considered a range of single-digit percent KC activations, however our "range" only had 3 entries. We have not simulated a 20% change of the number of activated synapse but we did simulate a slightly larger increase, by 25% (our Figure 5B). Our results show that the 25% increase leads to a substantial change in the somatic MBON voltage (Figure 5B,D). Given the nearly linear dependence of the voltage increase with the number of synapses (Figure 5I), we are confident that a 20% increase can be obtained with high precision from a linear interpolation based on the 25% increase (as can be any other increases within the range we cover with our three values for the number of synapses, i.e. 38 to 63 KCs, Figure 5). An increase from 5% to 6% of activated synapses therefore increases the voltage by about 80% of the value shown in Figure 5B, D which is a substantial difference. We would consider such a difference not a question of robustness but as a change that is likely physiologically relevant.

We have added a short discussion of this topic in the revised manuscript (line 340ff, at the very beginning of Section 2.4)

Line 99: Aso & Rubin 2016 belongs with citations in line 95.

Many thanks for pointing this out. We moved the citation to the correct place.

Line 122: "Drosophila" needs italics, throughout the manuscript.

All instances of "*Drosophila*" are now set in italics.

The authors devise a computational model of an MBON using a neuronal arbor reconstructed from volume electron microscopy by Takemura et al. 2017. That paper details that only 93% of all synapses were connected to an arbor, and only 86% of the synapses had known pre- and postsynaptic arbors. For the MBON that was used for modeling, what was the fraction of terminal ends labeled as uncertain, and where these clustered or scattered across the arbor?

The reviewer rightfully highlights the limitations of the EM-dataset that we used for our reconstruction. Indeed, Takemura and colleagues report that not all postsynaptic sites within the α-lobe could be assigned to identified cells. Importantly, however, the authors state that these synaptic sites very likely belong to identified cells within the dataset. Thus, for our dataset this would mean that the MBON itself may have up to 7 percent more input synapses but these synapses would belong to the identified 948 KCs. An individual KC may therefore at most have one or two additional input synapses, e.g. KCs currently annotated with 13 input synapses may form 14 input synapses. These differences would not significantly change any of the conclusions of our manuscript. We now addressed this point experimentally and analyzed the activation of KCs with 12, 13 and 14 input synapses and compared this to the activation of 13 random synapses (Figure 4F). We observed significant differences in MBON responses when changing the number of input synapses but not between KCs with 13 synapses and the activation of 13 random synapses. Thus, the number of input clearly changes postsynaptic responses, however, as we assume that these non-annotated synapses are likely uniformly distributed, these changes would not affect our physiological simulations of different sets of 50 KCs. Unfortunately, Takemura and colleagues did not specify the location of these non-annotated sites and we simply cannot re-trace the entire dendritic tree of the MBON. As they state however that "the small size and discontinuous nature of these neurites indicates that they are fragments of identified cells rather than collectively constituting an additional cell type", we assume that these sites were not clustered.

A similar rate of "tracing shortcomings" has been observed in the hemibrain dataset and in the current FlyWire approach. We decided that the most conservative approach would be to use the data as published as any modification, e.g. addition of 10% of synapses, would make later reproductions of our data more difficult. We now state the potential problems clearly in our paper (line 206ff, section 2.2).

Furthermore, the volume imaged with FIBSEM did not fully enclose the vertical lobe of the MB. Any estimate of what fraction of the chosen MBON's arbor is contained within the imaged volume? In other words, what analysis has been done here to ensure that the modeled arbor is representative of an MBON arbor in vivo, and what mitigating measures were taken to account for the potentially missing 14% or more of the arbor synapses and terminal dendrites?

Indeed the dataset of Takemura et al. did not include the entire vertical lobe of the MB. However, it included the entire α-lobe of the MB – only the α prime lobe is missing. As beautifully shown by Takemura et al., the entire dendritic tree of MBON-3alpha that only innervates the α-lobe is included and presented in Figure 2H and videos 4, 6 and 7 with the position of synaptic input sites. The striking visualization of Takemura and colleagues of this dendritic tree was one of the main reasons why we chose this MBON as an example for our combined in vivo and in silico analysis. While Takemura states that they could only precisely identify both pre- and postsynaptic partners for 86% of all synapses, this does not necessarily mean that 14% of KC to MBON synapses are missing in the dataset as e.g. only 7% of the postsynaptic sites remained unassigned (and these are the relevant ones for our MBON). We again carefully compared the annotations of KC to MBON-alpha3 synapses in Takemura et al. and Scheffer et al. (Hemibrain dataset) to evaluate the effects of a potential under-representation of synapses within the dendritic compartment of MBON-alpha3. Interestingly, the numbers for both MBON-alpha3 cells in the hemibrain dataset are below the numbers of Takemura et al. Specifically the numbers are for Takemura: MBON-alpha3A 12,770 synaptic connections from the 948 KCs (average 13.47 synapses), MBON-alpha3B 13,129 from 948 KCs (average 13.85 synapses) vs Scheffler: MBON-alpha3A 11,950 (+1610 which are not classified = 13,560) synapses from 894 KCs (average 12.44 synapses), MBON-alpha3B 11,121 (+ 1,572 not classified = 12,693) synapses from from 897 KCs (average 13.32 synapses).

These numbers are highly similar between the two data sets and also between the two different MBON-alpha3 neurons. Importantly however, the numbers are lower in the Hemibrain data set. Thus, the variation between individual flies (or data sets) seems to be in the range of 10%. As such, the Takemura dataset likely represents a very good approximation despite some shortcomings due to either the EM fixation (see below) or due to annotation problems. As we did not want to introduce any arbitrary mistakes in our analysis we chose to adhere to the numbers provided in the Takemura data set. We now clearly state the potential shortcomings of using a single EM data set as the basis for a simulation (line 206ff). We still think that this is currently the best possible approach.

The authors report using the 10XUAS-IVS-mCD8::GFP to label the MBON, so that they can then record electrophysiologically with patch-clamp. What is the effect of inserting so many mCD8 proteins (a large transmembrane protein) into the neuron's membrane on the voltage potential and action potential formation and transport? The 10XUAS is particularly strong. How does the morphology of the imaged neuron differ from that of the EM-reconstructed neuron, regarding calibers and amount of cable? For this purpose, a cytosolic GFP targeting the soma or nucleus and poorly diffusing into the arbor would have been far preferable, as the effect of inserting transmembrane proteins in neurons' membranes on resting potentials is well reported.

We would like to thank the reviewer for this suggestion. We now performed additional recordings with a cytosolic GFP construct. The data from 4 additional recordings is now presented in Figure 1 Supplement 2. In this new Figure we now provide a direct comparison between the mCD8-GFP based recordings and the cytosolic GFP based recordings and present all raw data traces. This analysis demonstrated that the averaged traces of the new recordings fall within the data range of our initial dataset (Figure 1 Supplement 2 I). Thus, the source of GFP used to label these cells did not have any significant consequences for the recordings. In addition, with this new data set we could clearly demonstrate that our electrophysiological data are reproducible and that the original data set provides accurate values for our computer simulations.

Line 155: average resting potential for the MBON is reported at -56.7 mV +/- 2.0 mV. In Hige et al. 2015a, cells were held at -70 or -60 mV. Nowhere does Hige et al. 2015a report on the resting potential.

The average resting membrane potential of the recorded MBONs was at -56.7 mV +/- 2.0 mV. To our knowledge, we are the first to report precise electrophysiological values for this particular cell type. It is correct that Hige et al., 2015a do not report values however, it is possible to estimate these values from the raw traces provided in their publication. Importantly, our value of approximately -60 mV is in line with data from a number of central neurons in *Drosophila* (e.g. Gu and O´Dowd, 2006; Wilson et al., 2004; Groschner et al., 2022). We now state more appropriately (line 154ff): This value for the restring membrane potential is in line with prior measurements of MBONs in vivo and of other central neurons in *Drosophila* (Hige et al., 2015a, Gu and O´Dowd, 2006; Wilson et al., 2004; Groschner et al., 2022)

Line 174: amplitude of action potentials was rather small, but in Hige et al. 2015a action potentials typically exceeded 200 pA. Is this what the authors mean by small? Just how small were the recorded action potentials?

The amplitude of the action potentials (4.3 *±* 0.029 mV) represent relatively low values compared to previously published data from different cell types in the central brain of the adult fruit fly (see references above). Even between different MBONs a high variability in action potential amplitudes has been observed (Hige et al., 2015a). While precise values were not reported in this publication, based on the traces the values range between very small (gamma2), 10 mV (alpha2sc) and 25 mV (gamma1). The small amplitudes observed for MBON alpha3 are likely a consequence of the unipolar morphology of the neurons with a very long neurite connecting the dendritic input region to the cell soma. This morphology is especially pronounced for MBON alpha3 (see Figure 1A).

Line 176: by small amplitudes and the explanation on the long neurite connecting the dendritic arbor with the soma, you mean that the signal is attenuated over long distances?

Yes, due to the very long neurite the signal is attenuated significantly.

Line 179: how was the membrane capacitance calculated?

Membrane capacitance was determined by dividing tau by the membrane resistance (Cm = tau/Rm). We now state this in the text (line 179).

Table 1: 5 neurons were used. How do you know they are all MBON-α3? Has it been confirmed that the GAL4 line doesn't have stochastic expression among similar yet different sibling neurons of the same lineage? How many other MBONs innervate the tip of the α lobe and do any of them share neuroblast lineage with MBON-α3? The large differences in measured values listed in Table 1 could be explained by having measured similar yet different neurons. Did you run a battery of tests before and after the measurements to ensure the recorded neurons remained in good health throughout the measurement session? Such tests often consist in a ramp of current injections and the recording of the neuron's responses, which are then compared between before and after the experimental measurements of membrane properties (like the current step protocol of Figure 1F).

The MB082C split-Gal4 line we used for our recordings is a highly specific driver line which is only expressed in the two MBON-alpha3 neurons per hemisphere across all observed preparations. These neurons do not share any overlap (position of dendrites plus position of soma) with any other MBON. Thus we are certain that all our recordings were performed on MBON-alpha3 neurons. However, the two MBON-alpha3 neurons innervating the same position in the α lobe are indistinguishable from each other. As there is no possibility to differentiate between the two and as their morphological features are almost identical (see numbers in our answer to the EM dataset above) we treated them simply as MBON-alpha3. We now state this clearly in our manuscript. As we are certain about the identity of the MBON, any differences likely result from inter-individual biological differences.

Line 184: why only 3 cells? In Drosophila, recording from e.g. 10 cells, all homologous cells across 10 individuals, gives e.g., 8 responding with excitation to a sensory stimuli with some variation and 2 responding with inhibition. There is a lot of variability in the responses. Recording from only 3 cells seems risky, statistically speaking. What justifies this low number?

We agree. We now added 4 additional recordings using a cytoplasmic GFP as a labelling source to the manuscript (Figure 1 Supplement 2, Table 1 Source data 1). These values of these recordings fall in between the values of our prior recordings and thus validate our prior results. We now have 7 detailed recordings for these cells that likely show the range of biological variation. It is technically extremely challenging to perform these recordings.

Line 186: "To to further".

We corrected this mistake.

Line 199: when you say that the measurements are in "good agreement with prior recordings" of other neurons in Drosophila, what do you mean exactly? How similar, how far off, by what parameters?

We revised this section. We now simply report the results of our recordings and state that MBONalpha3 is a spike-frequency adapting neuron like other MBONs. We removed the prior sentence with comparisons to other neurons.

Line 203: might as well mention that there were 948 reconstructed KCs synapting onto MBON-α3, so 5% is 50. Spare the reader remembering where the 50 was picked from. (If you correct this to 6%, would be 57 KCs). And you seem to not keep in mind that the KCs responding to a specific odor may be correlated in their synaptic connectivity strength onto MBON-α3. Data to this end may be included in Li et al. 2020 eLife where the whole mushroom body is reconstructed, including the olfactory projection neurons, so such correlations if any may be evident in that data set.

We now state directly that the number 50 is deduced from the 5% of KCs at the appropriate position in the text.

The reviewer is certainly correct that there is now clear evidence that the PN>KC input is not entirely random. Importantly, Li et al. demonstrated that inputs from different sensory modalities are especially segregated – however, MBON-alpha3/14 receives almost exclusively olfactory input (>97% as deduced from Figure 15, Supplement 2). In their models Li et al., still find some evidence for non-random sampling at the PN>KC synapse but this was modest at best and they observed only minor persistance to the KC>MBON level. In addition Zheng et al., 2022 demonstrate that some specific odors may be overrepresented compared to others to enable preferential encoding of naturally meaningful stimuli. In our first analysis presented here we decided to focus on the robustness of encoding principles by assuming random connectivity. We certainly aim to pursue a targeted analysis in a future study focussing on multiple MBONs.

Line 210: a complete reconstruction of MBON-α3 now exists, from either the FAFB volume or the Hemibrain volume. In the methods you mention you used the Hemibrain data set for the axon.

In the FAFB volume our MBON is not fully reconstructed. The reconstructions from Takemura and the Hemibrain are very similar and consistent regarding the number of KC>MBON synapses (see above). As we started our first analysis before the publication of the Hemibrain focusing only on the dendritic tree (Hafez et al., 2019, bioRxiv) we build our current study on this analysis and complemented the morphology reconstruction with data from the Hemibrain.

Line 209: the 12,770 synaptic connections aren't "all", these are the ones reported from the anatomical reconstruction from volume electron microscopy. According to the source papers (Takemura et al. 2017) about 10% of all synapses are missing. An analysis of how these missing synapses impacts the structure of the arbor is absent from the paper.

Please see our responses above to this topic. We now highlight these issues in the results when first describing our approach (line 206ff).

In addition, sample preparation for electron microscopy with chemical fixation alters the fine anatomical details, including the length of terminal dendrites and the calibers of neurites throughout. See e.g., Korogod et al. 2015 eLife "Ultrastructural analysis of adult mouse neocortex comparing aldehyde perfusion with cryo fixation" and the follow up paper Tamada et al. 2020 eLife "Ultrastructural comparison of dendritic spine morphology preserved with cryo and chemical fixation". What measures were taken to correct or mitigate these artifactual differences with in vivo neurons?

Korogod et al. showed that the main effect of fixating somatosensory mouse cortex using the standard aldehyde method was a substantial decrease of the extracellular space and an increase of astrocytic volume, compared to a ’fresh’ (not fixated) preparation. This was not the case, however, for neurites. They found that "the volume fraction occupied by axons and dendrites was similar between the fixation conditions" (their p 3).

There was also no indication of systematic loss or erroneous addition of synapses (the shape of synaptic vesicles was modified but we do not make use of this feature in our study). Synaptic density in the chemically fixed neuropil was 38% higher than cryo-fixated neuropil, i.e. they measured 38% more synapses per volume. This is slightly lower than the total tissue shrinkage they observed (1/0.7 *≈* 43%), a difference they explained by the fact that there are no synapses in structures like cell bodies and blood vessels. It seems therefore likely that the increase of the synapse density is primarily due to a decrease in the denominator (the volume) due to the loss of recorded extracellular space, rather than in the numerator (number of synapses).

Given that the main effect of chemical fixation is a sharp reduction of extra-cellular space, with no significant changes of neurite dimensions, we feel that the EM data we use are at least a good first approximation of the physiological state. A previous study of *Drosophila* neurons from Rachel Wilson’s lab (Tobin et al. 2017, https://doi.org/10.7554/*eLife*.24838) came to a similar conclusion.

Tamada et al. focused on fixation-induced morphology changes of dendritic spines, again in mouse cerebral cortex. They found that the length of the spines as well as the spine head volume was not significantly changed by traditional (chemical) fixation preceding EM. They did find that chemical fixation resulted in significantly (by 42%) thicker spine necks, however. Assuming identical cytoplasmatic specific resistivity, using the measured (per chemical fixation) spine neck diameter would result in significantly lower resistance between spine head and dendrite than the actual physiological value.

Spine-like structures have been observed in the *Drosophila* nervous system (e.g. Leiss et al. "Characterization of dendritic spines in the *Drosophila* central nervous system." Developmental neurobiology 69.4 (2009): 221-234). However, the relevance of these results for our model is unknown. In the published data no spine-like structures or locations on the dendritic tree of MBON-*α*3 have been characterized; so we are not able to evaluate details of the impact that different diameters of the spine necks would have. In the absence of such information, we can not address (potential) differences in the resistance between individual synaptic contacts and the dendrites. Instead, we fit a common value for the cytoplasmatic resistivity for the whole neuron (our Table 2). We agree that it would, indeed, be very interesting to study the effect of synapses on spines (for those neurons where they exist) compared to that of synapses located directly on the dendritic surface, but this requires the availability of data characterizing location and, ideally, geometry of dendritic spines on neurons of interest. Due to these reasons we did not include any correction factor in our analysis.

Later, Figure 2F strongly supports the appropriateness of the model, yet, the above points merit discussion and even exploration: how much of the dendritic arbor can you miss and still get the same result? What does the response to current injection depend on, cable, number of synapses, synapse spatial location, cable calibers, tapering of cable? What cable truncations are tolerable? This is very important information towards future computational studies based on neuronal morphologies reconstructed from volume electron microscopy.

We agree that these are important points. It is important to note that we provide results for our simulations at two different points. First, at the proximate neurite, the "exit" point of the dendritic tree and potential site of action potential initiation, and second at the soma region. Importantly, the region of the proximate neurite is still within the EM-dataset of Takemura et al. – thus, no relevant dendritc arbor sections are missing and no assumptions regarding the diameter of any neuronal structures have been made. We provide a direct comparison to the soma part that is based on a different dataset and approximations as it is currently not included in any EM dataset (the upcoming FlyWire dataset will hopefully resolve this issue). Our comparison demonstrates very few differences between these two points of analysis indicating that our dataset is very robust – but of course the main computation occurs within the dendritic compartment that was not modified in our analysis (see also responses above for our reasoning not to perform any corrections). We therefore cannot draw any significant conclusions regarding physical alterations of the dendritic compartment and would like to stick to our "conservative" approach for this initial study. We will certainly test more challenging models in future studies.

Figure 2 legend: what is the evidence that the "proximal neurite" in green in Figure 2B is the site of axon potential generation? Gouwens and Wilson 2009 pointed at a region anywhere between the root of the dendritic tree and half-way through the axon of the uniglomerular olfactory projection neuron they modeled.

We currently do not have any evidence that this is the position of action potential initiation. Based on immunohistochemical analyses of the *Drosophila* para channel in other neurons the position is likely in close proximity to the dendritic arbor. Here, we used this position for our comparative analysis with the soma as this region was still confined within the Takemura EM reconstruction – we state this now more clearly in the revised manuscript in the legend to Figure 2. As we observed consistent results between the two regions it would not make a significant difference if this region would be moved. In future work we hope to precisely determine the site of action potential generation using genetic methods.

Does the site of axon potential generation emerge from your model, or did you specify it in the model?

Please see above, we specified the site in the model.

Why is the two-tailed non-parametric Spearman correlation the correct statistic to compare the modeled and the experimentally measured membrane potential in Figure 2F?

Many thanks for highlighting this. This comparison is not necessary as the fitting was done in NEURON using the mean squared error. We now state this appropriately in the Methods section.

Figure 2 legend reads "see appendix" but there isn't any appendix to the manuscript?

We changed this and now only refer to the figure supplement.

Line 249: "the average number of synaptic contacts from a KC to this MBON is 13.47". This statement ought to be qualified: for the single MBON-α3 measured in Takemura et al. 2017, and with the caveat of ~10% of synapses potentially missing. You could just as easily apply a correction factor and say the average number is about 14.7 + 1.47 = 16.2. Would this change the outcome of your model?

Please see our discussion of this issue above. We now clearly state that 10% of synapses potentially missing at the beginning of the section and provide our reasoning. We also now tested the responses to 12, 13 or 14 synapses in Figure 4F.

Please don't use "PN" as an acronym for proximal neurite. First, eLife doesn't restrict the length of your test. Second, PN is an established acronym, universally across all neuroscience literature, for projection neuron. Plus, the "proximal neurite" (as per figure 2B) might as well be called the putative AIS (axon initial segment; pAIS for "putative") where the integration of inputs across the entire dendritic tree take place and the axon potential is initiated.

Many thanks for pointing this out. We now use PN as an acronym for projection neuron and do not abbreviate proximal neurite. We now also highlight that this is the likely site of action potential initiation. However, as this is not based on experimental evidence we would like to remain conservative and refer to it only as the proximal neurite.

Figure 3H: in the measurement of "local dendritic section volume", did you correct for volume artifacts induced by using (in purpose!) an incorrect osmolarity of the buffers when fixing the tissue in the sample preparation protocol for electron microscopy? See Korogod et al. 2015 eLife and Tamada et al. 2020 eLife.

Please see our response to potential fixation artifacts above.

Line 291: "this value is in good agreement with in vivo data for MBONs". Please could you specify what this agreement is, how close, some details.

Many thanks for pointing this out. We now removed this sentence as we only deduced this value from odor evoked activations in vivo that activate approximately 5% of KCs and result in action potential generation. Thus a direct comparison is currently not possible.

Line 293: in line with analyzing all KCs with exactly 13 synaptic inputs onto MBON-α3, what's the result of analyzing the voltage excursion from drawing random subsets of 13 synapses? (or 16 as per the correction, see above). Are the natural groups of 13 synapses different in their effect on the neuron's voltage than artificial groupings?

Many thanks for suggesting this very interesting experiment. We now performed the experiment and provide a comparison between KCs with exactly 12, 13 and 14 synapses to a random set of 13 synapses (Figure 4F). This experiment demonstrated no significant difference between the activation of KCs with 13 synapses and 13 random synapses. Interestingly though, slightly more variation could be observed in the dataset of KCs with 13 synapses. This may indicate some biological relevance however, the differences compared to activating one additional synapse are negligible in line with our model.

Line 299: inaccurate statement: "Given that ≈ 5% of the 984 KCs innervating MBON-α3 are typically activated by an odor". Instead, what is known from the literature is that, given the presence of the GABAergic neuron APL in the mushroom body which acts as the inhibitory unit of a winner-take-all configuration, only 6% (not 5%) of KCs simultaneously respond to any one odor. Plus when the APL neuron is inhibited, a huge double-digit percent of KCs are active in response to an odor.

We now corrected this statement and highlight that only 5-6% (range 3-9%) of neurons reliably respond to individual odors. This is the value that is likely relevant for memory association. As we do not address the inhibitory role of the APL neuron in our current study we would like keep our study focused and potentially explore these issues in a future study.

Line 311: there is now far better evidence of stochastic odor encoding by KCs than Caron et al. 2013. See Zheng e t al. 2020 bioRxiv "Structured sampling of olfactory input by the fly mushroom body", and Li et al. 2020 eLife, and also, for larvae, Eichler et al. 2017 Nature.

We are sorry for our oversight and now include these citations.

Line 319: see also Baltruschat L, Prisco L, Ranft P, Lauritzen JS, Fiala A, Bock DD, Tavosanis G. Circuit reorganization in the *Drosophila* mushroom body calyx accompanies memory consolidation. Cell reports. 2021 Mar 16;34(11):108871.

We added this citation.

Line 337: the differences in the somatic amplitudes may be significant statistically, but are they meaningful? In other words, the effect size looks like near zero. The real, and important difference, is in Line 345 where it is stated that "we observed some differences in the slope of the responses between the different tuning modalities (Figure 5J,K,L)."

We agree that the observed differences may not be biologically relevant but we can not formally exclude this possibility at the moment. We now highlight the differences in the slopes of the responses further in the discussion.

Line 350: Scheffer et al. 2020 is not an appropriate citation for the statement "Te ability of an animal to adapt its behavior to a large spectrum of sensory information requires specialized neuronal circuit motifs". Rather, a textbook such as Kandel et al. Principles of Neural Science, or no reference at all, would be appropriate. You could also delete the sentence without loss.

We agree and deleted the sentence.

Line 353: *Drosophila* must go in italics, it's a species name. Multiple occurrences throughout.

As mentioned above, "*Drosophila*" is now set in italics throughout.

Line 356: through both short-term and long-term memory. A good example is Aso & Rubin 2016 eLife.

Many thanks for this useful suggestion – we adjusted the text accordingly.

Line 359: note the olfactory system of the fly has a sort of "fovea", Zheng et al. 2020 bioRxiv.

As the study is now peer-reviewed and published we included it in the reference list and altered the statement to reflect the fact that processing of odor information does not exclusively depend on stochastic connectivity (line 65ff) – in addition we refer to this paper in our discussion of the two possible plasticity modes (line 504ff).

Line 362: this sentence needs work, I am not sure what it means: "Individual flies display idiosyncratic, apparently random connectivity patterns that transmit information of specific odors to the output circuit of the MB."

We agree and deleted the sentence.

Line 366: MBONs can only each be classified as approach or avoidance within specific behavioral paradigms. In different contexts, including different physiological states (e.g., hunger, satiation, others), the classification changes.

Yes, we now state this more appropriately in the revised version of the manuscript (line 409ff).

Line 381: prior work includes Tobin et al. 2017 eLife, where EM-reconstructed dendrites of olfactory projection neurons were modeled to understand the impact of dendritic arbor size on neuronal function.

Many thanks for pointing this out. We now cite this work.

Line 386: again, these numbers aren't precise. There's about 10% of missing synapses to consider, and potentially additional Kenyon cells. And some of the KCs, particularly those with low number of synaptic connections to MBON-α3, may have been connected in error.

We now highlight this point in the discussion and state (line 432ff): "While this EM reconstruction may contain some mistakes in synaptic connectivity as e.g. up to 10% of synaptic sites remained unassigned within the dataset (Takemura et al., 2017), it currently represents the best possible template for an *in silico* reconstruction."

Line 397: Strongest finding of this work: "The location of an individual synaptic input within the dendritic tree has therefore only a minor effect on the amplitude of the neuron's output, despite large variations of local dendritic potential." Would be best to surface it more.

We agree. We think that the importance of this finding is appropriately highlighted in the current form.

Line 402: a comparison would be appropriate with neurons from the crayfish stomatogastric ganglion (STG) as described by Eve Marder lab's, with published findings such as neurons being electrotonically compact despite their large size in mature adult animals. For example, Otopalik et al. 2017 eLife "When complex neuronal structures may not matter", where the authors "quantify animal-to-animal variability in cable lengths (CV = 0.4) and branching patterns in the Gastric Mill (GM) neuron". And also Otopalik et al. 2019 eLife "Neuronal morphologies built for reliable physiology in a rhythmic motor circuit".

Many thanks for pointing this out. We now include this work in our discussion and cite these papers.

Line 406: you forgot to cite Jackie Schiller's work on cortical pyramidal cells and their tuft dendrites, some of which predates all the cited work in this statement.

We regret this oversight and now appropriately cite the work.

Line 412: your model, as per Figure 2F, rather very closely matches the observed electrophysiological responses of the neuron under study. In what further ways could you model more closely match experimental observations? This would be very instructive to the reader.

This may be a misunderstanding. We simply highlight the possibility to incorporate active currents within the dendritic compartment if these should be discovered in *Drosophila* in the future. We now make this more clear by starting the sentence with "In case…"

Line 415: by ensembles of both KCS and MBONs, not just KCs, if you are including the memory part and not only the input representation part.

Yes, we changed the sentence accordingly.

Line 416: for most of the paper, you quoted these papers to justify the 5% of KCs being simultaneously active in response to an odor. Now the range is shown as 3 to 9%. This discrepancy ought to be reconciled.

As stated above we now make this point clear throughout the paper and discuss it more carefully (line 340ff).

Line 422: again, nowhere in the manuscript so far did you detail in what way your simulation and experimental findings match those of prior experimental reports regarding neuron response and physiological properties.

We agree that the sentence was misleading. We now compare this activation to the observed odor evoked changes in action potential frequency in MBONs in the studies by Hige and colleagues (line 469ff).

Line 425: this statement is inaccurate. An individual KC does not encode for a single odor. That would almost never be the case even if a KC was single-claw, as in, exclusively integrated inputs from a single projection neuron: individual projection neurons rarely encode an odor; it's the population of projection neurons that does. Similarly, ensembles of co-active KCs together represent an odor, and do so more narrowly and accurately than the population of projection neurons that excited them.

This seems to be a misunderstanding. In line 425 of the original version we did not talk about individual KCs but about sets of 50 KCs. This is in line with the statement of the reviewer.

Line 452: left unresolved remains the question of why would both mechanisms exist, as in, is the combination of altering the KC-MBON synapse and altering the PN-KC synapse better in some dimension than altering either alone? Does this relate perhaps to a possible dynamic nature of the olfactory "fovea" proposed by Zheng et al. 2020 bioRxiv as presumably static?

We now extended this part of the discussion and speculate about the relevance of these different plasticity modes (line 504ff).

Line 466: what is standard fly food? Please define it. Choice of food affects very much behavioral assays, for example.

As we did not perform any behavior assay, we did not provide a detailed recipe. We basically follow BDSC with small modifications.

Line 476: no antigen saturation steps in the immunohistochemistry protocol? Please revise.

Yes, we did not perform any antigen saturation steps as this is not necessary for our anti-GFP stainings.

Line 505: what were the settings of the puller? These would be necessary to reproduce your glass capillaries. Did you grind the tips, and if so, how?

We now provide the detailed settings of the puller.

Line 534: how were the R_series, R_input and R_m determined with Igor Pro?

We now specify this in the Methods section.

Line 561: why was a threshold of 600 MΩ used to exclude cells? How many cells were recorded in total, and how many were excluded?

We now specified the inclusion criteria more carefully. Cells not matching those criteria were not included in any analysis.

Line 584: once again the data is the best human effort in proofreading a semiautomatic segmentation. It is not the absolute truth. Also, each individual fly is somewhat different, so this is merely one fairly complete yet partial reconstruction, and not assured to be error free, particularly of errors of omission, of a single MBON from a single individual. Would be appropriate to remark this clearly.

We now state this clearly throughout the manuscript and again explicitly in the methods section.

Did you provide the NeuroML or similar files necessary to run the model in the NEURON simulation software? These should be appended to the manuscript as supplemental data.

Yes, all code necessary to run the model in the NEURON simulation software is included and documented in the Dataverse deposit, doi.org/10.7281/T1/HRK27V, as listed under ("Availability of data, materials, and code").

Line 596: if "parameters were tuned until the computed voltage excursion at the soma matched our electrophysiological recordings", what is the rationale for comparing the voltage potential of the simulation with those of the experimental observations?

The described part in the methods refers to the establishment of the basic electrophysiological properties of the simulated neuron. Our analysis then address the activation of this neuron.

Line 607: where do these ranges of physiologically plausible values come from? Citation, or measurements done in the lab and therefore a figure is needed to show them? Are these ranges from the literature listed in Table 2? Likely yes, would be appropriate to cite them here too or at least explicitly point to Table 2.

We now point to Table 2 in this section.

Line 616: innervation of MBONs is cholinergic because KCs are cholinergic, but that's not from Takemura et al. 2017, instead from Barnsted et al. 2016 Neuron.

We adjusted it accordingly.

Line 622: would be appropriate to include the python scripts used to configure and run the NEURON simulation as supplemental material. Likewise for the matlab scripts used to load the analyzed data and plot it. The raw data ought to be included as CSV files or similar.

All code and raw data necessary to run the model in the NEURON simulation software and to recapitulate our work is included and documented in the Dataverse deposit, doi.org/10.7281/T1/HRK27V, as listed under ("Availability of data, materials, and code").

Reviewer #2 (Recommendations for the authors):"The cellular architecture of memory modules in *Drosophila* supports stochastic input integration" is a classical biophysical compartmental modelling study. It takes advantage of some simple current injection protocols in a massively complex mushroom body neuron called MBON-a3 and compartmental models that simulate the electrophysiological behaviour given a detailed description of the anatomical extent of its neurites.This work is interesting in a number of ways:– The input structure information comes from EM data (Kenyon cells) although this is not discussed much in the paper– The paper predicts a potentially novel normalization of the throughput of KC inputs at the level of the proximal dendrite and soma– It claims a new computational principle in dendrites, this didn't become very clear to meProblems I see:– The current injections did not last long enough to reach steady state (e.g. Figure 1FG), and the model current injection traces have two time constants but the data only one (Figure 2DF). This does not make me very confident in the results and conclusions.

These are two important but separate questions that we would like to address in turn.

As for the first, in our new recordings using cytoplasmic GFP to identify MBON-alpha3, we performed both a 200 ms current injection and performed prolonged recordings of 400 ms to reach steady state (for all 4 new cells 1’-4’). For comparison with the original dataset we mainly present the raw traces for 200 ms recordings in Figure 1 Supplement 2. In addition, we now provide a direct comparison of these recordings (200 ms versus 400 ms) and did not observe significant differences in tau between these data (Figure 1 Supplement 2 K). This comparison illustrates that the 200 ms current injection reaches a maximum voltage deflection that is close to the steady state level of the prolonged protocol. Importantly, the critical parameter (tau) did not change between these datasets.

Regarding the second question, the two different time constants, we thank the reviewer for pointing this out. Indeed, while the simulated voltage follows an approximately exponential decay which is, by design, essentially identical to the measured value (*τ≈* 16ms, from Table 1; see Figure 1 Supplement 2 for details), the voltage decays and rises much faster immediately following the onset and offset of the current injections. We believe that this is due to the morphology of this neuron. Current injection, and voltage recordings, are at the soma which is connected to the remainder of the neuron by a long and thin neurite. This ’remainder’ is, of course, in linear size, volume and surface (membrane) area much larger than the soma, see Figure 2A. As a result, a current injection will first quickly charge up the membrane of the soma, resulting in the initial fast voltage changes seen in Figure 2D,F, before the membrane in the remainder of the cell is charged, with the cell’s time constant *τ*.

**Author response image 1. sa2fig1:** Simplified circuit of a small soma (left parallel RC circuit) and the much larger remainder of a cell (right parallel RC circuit) connected by a neurite (right 100M*Ω* resistor). A current source (far left) injects constant current into the soma through the left 100M*Ω* resistor.

We confirmed this intuition by running various simplified simulations in *Neuron* which indeed show a much more rapid change at step changes in injected current than over the long-term. Indeed, we found that the pattern even appears in the simplest possible two-compartment version of the neuron’s equivalent circuit which we solved in an all-purpose numerical simulator of electrical circuitry.

**Author response image 2. sa2fig2:** Somatic voltage in the circuit in Figure 1 with current injection for about 4. 5ms, followed by zero current injection for another *≈* 3.5ms..

**Author response image 3. sa2fig3:** : Somatic voltage in the circuit, as in Figure 2 but with current injected for approx. 15ms.

(https://www.falstad.com/circuit). The circuit is shown in Author response image 1. We chose rather generic values for the circuit components, with the constraints that the cell capacitance, chosen as 15pF, and membrane resistance, chosen as 1G*Ω*, are in the range of the observed data (as is, consequently, its time constant which is 15ms with these choices); see Table 1 of the manuscript. We chose the capacitance of the soma as 1.5pF, making the time constant of the soma (1.5ms) an order of magnitude shorter than that of the cell.

Author response image 2 shows the somatic voltage in this circuit (i.e., at the upper terminal of the 1.5pF capacitor) while a -10pA current is injected for about 4.5ms, after which the current is set back to zero. The combination of initial rapid change, followed by a gradual change with a time constant of *≈* 15ms is visible at both onset and offset of the current injection. Author response image 3 show the voltage traces plotted for a duration of approximately one time constant, and Author response image 4 shows the detailed shape right after current onset.

While we did not try to quantitatively assess the deviation from a single-exponential shape of the voltage in Figure 2E, a more rapid increase at the onset and offset of the current injection is clearly visible in Author response image 4. This deviation from a single exponential is smaller than what we see in the simulation (both in Figure 2D of the manuscript, and in the results of the simplified circuit in Author response image 4). We believe that the effect is smaller in Figure E because it shows the average over many traces. It is much more visible in the ’raw’ (not averaged) traces. Two randomly selected traces from the first of the recorded neurons are shown in Figure 2 Supplement 2 C. While the non-averaged traces are plagued by artifacts and noise, the rapid voltage changes are visible essentially at all onsets and offsets of the current injection.

**Author response image 4. sa2fig4:** Somatic voltage in the circuit, as in Figure 2 but showing only for the time right after current onset, about 2. 3ms.

We have added a short discussion of this at the end of Section 2.3 to briefly point out this observation and its explanation. We there also refer to the simplified circuit simulation and comparison with raw voltage traces which is now shown in the new Figure 2 Supplement 2.

– The time constant in Table 1 is much shorter than in Figure 1FG?

No, these values are in agreement. To facilitate the comparison we now include a graphical measurement of tau from our traces in Figure 1 Supplement 2 J.

– Related to this, the capacitance values are very low maybe this can be explained by the model's wrong assumption of tau?

Indeed, the measured time constants are somewhat lower than what might be expected. We believe that this is because after a step change of the injected current, an initial rapid voltage change occurs in the soma, where the recordings are taken. The measured time constant is a combination of the ’actual’ time constant of the cell and the ’somatic’ (very short) time constant of the soma. Please see our explanations above.

Importantly, the value for tau from Table 1 is not used explicitly in the model as the parameters used in our simulation are determined by optimal fits of the simulated voltage curves to experimentally obtained data.

– That latter in turn could be because of either space clamp issues in this hugely complex cell or bad model predictions due to incomplete reconstructions, bad match between morphology and electrophysiology (both are from different datasets?), or unknown ion channels that produce non-linear behaviour during the current injections.

Please see our detailed discussion above. Furthermore, we now provide additional recordings using cytoplasmic GFP as a marker for the identification of MBON-alpha3 and confirm our findings. We agree that space-clamp issues could interfere with our recordings in such a complex cell. However, our approach using electrophysiological data should still be superior to any other approach (picking text book values). As we injected negative currents for our analysis at least voltage-gated ion channels should not influence our recordings.

- The PRAXIS method in NEURON seems too ad hoc. Passive properties of a neuron should probably rather be explored in parameter scans.

We are a bit at a loss of what is meant by the PRAXIS method being "too ad hoc." The PRAXIS method is essentially a conjugate gradient optimization algorithm (since no explicit derivatives are available, it makes the assumption that the objective function is quadratic). This seems to us a systematic way of doing a parameter scan, and the procedure has been used in other related models, e.g. the cited Gouwens & Wilson (2009) study.

Questions I have:– Computational aspects were previously addressed by e.g. Larry Abbott and Gilles Laurent (sparse coding), how do the findings here distinguish themselves from this work.

In contrast to the work by Abbott and Laurent that addressed the principal relevance and suitability of sparse and random coding for the encoding of sensory information in decision making, here we address the cellular and computational mechanisms that an individual node (KC*>*MBON) play within the circuitry. As we use functional and morphological relevant data this study builds upon the prior work but significantly extends the general models to a specific case. We think this is essential for the further exploration of the topic.

– What is valence information?

Valence information is the information whether a stimulus is good (positive valence, e.g. sugar in appetitive memory paradigms, or negative valence in aversive olfactory conditioning – the electric shock). Valence information is provided by the dopaminergic system. Dopaminergic neurons are in direct contact with the KC>MBON circuitry and modify its synaptic connectivity when olfactory information is paired with a positive or negative stimulus.

– It seems that Martin Nawrot’s work would be relevant to this work.

We are aware of the work by the Nawrot group that provided important insights into the processing of information within the olfactory mushroom body circuitry. We now highlight some of his work. His recent work will certainly be relevant for our future studies when we try to extend our work from an individual cell to networks.

– Compactification and democratization could be related to other work like Otopalik et al. 2017 eLife but also passive normalization. The equal efficiency in line 427 reminds me of dendritic/synaptic democracy and dendritic constancy.

Many thanks for pointing this out. This is in line with the comments from reviewer 1 and we now highlight these papers in the relevant paragraph in the discussion (line 442ff).

– The morphology does not obviously seem compact, how unusual would it be that such a complex dendrite is so compact?

We should have been more careful in our terminology, making clear that when we write ’compact’ we always mean ’electrotonically compact," in the sense that the physical dimensions of the neuron are small compared to its characteristic electrotonic length (usually called *λ*). The degree of a dendritic structure being electrotonically compact is determined by the interaction of morphology, size and conductances (across the membrane and along the neurites). We don’t believe that one of these factors alone (e.g. morphology) is sufficient to characterize the electrical properties of a dendritic tree. We have now clarified this in the relevant section.

– What were the advantages of using the EM circuit?

The purpose of our study is to provide a "realistic" model of a KC>MBON node within the memory circuitry. We started our simulations with random synaptic locations but wondered whether such a stochastic model is correct, or whether taking into account the detailed locations and numbers of synaptic connections of individual KCs would make a difference to the computation. Therefore we repeated the simulations using the EM data. We now address the point between random vs realistic synaptic connectivity in Figure 4F. We do not observe a significant difference but this may become more relevant in future studies if we compute the interplay between MBONs activated by overlapping sets of KCs. We simply think that utilizing the EM data gets us one step closer to realistic models.

– Isn't Figure 4E rather trivial if the cell is compact?

We believe this figure is a visually striking illustration that shows how electrotonically compact the cell is. Such a finding may be trivial in retrospect, once the data is visualized, but we believe it provides a very intuitive description of the cell behavior.

Overall, I am worried that the passive modelling study of the MBON-a3 does not provide enough evidence to explain the electrophysiological behaviour of the cell and to make accurate predictions of the cell's responses to a variety of stochastic KC inputs.

In our view our model adequately describes the behavior of the MBON with the most minimal (passive) model. Our approach tries to make the least assumptions about the electrophysiological properties of the cell. We think that based on the current knowledge our approach is the best possible approach as thus far no active components within the dendritic or axonal compartments of *Drosophila* MBONs have been described. As such, our model describes the current status which explains the behavior of the cell very well. We aim to refine this model in the future if experimental evidence requires such adaptations.

Reviewer #3 (Recommendations for the authors):This manuscript presents an analysis of the cellular integration properties of a specific mushroom body output neuron, MBON-α3, using a combination of patch clamp recordings and data from electron microscopy. The study demonstrates that the neuron is electrotonically compact permitting linear integration of synaptic input from Kenyon cells that represent odor identity.Strengths of the manuscript:1) The study integrates morphological data about MBON-α3 along with parameters derived from electrophysiological measurements to build a detailed model.2) The modeling provides support for existing models of how olfactory memory is related to integration at the MBON.Weaknesses of the manuscript:1) The study does not provide experimental validation of the results of the computational model.

The goal of our study is to use computational approaches to provide insights into the computation of the MBON as part of the olfactory memory circuitry. Our data is in agreement with the current model of the circuitry. Our study therefore forms the basis for future experimental studies; those would however go beyond the scope of the current work.

2) The conclusion of the modeling analysis is that the neuron integrates synaptic inputs almost completely linearly. All the subsequent analyses are straightforward consequences of this result.

We do, indeed, find that synaptic integration in this neuron is almost completely linear. We demonstrate that this result holds in a variety of different ways. All analyses in the study serve this purpose. These results are in line with the findings by Hige and Turner (2013) who demonstrated that also synaptic integration at PN>KC synapses is highly linear. As such our data points to a feature conservation to the next node of this circuit.

3) The manuscript does not provide much explanation or intuition as to why this linear conclusion holds.

We respectfully disagree. We demonstrate that this linear integration is a combination of the size of the cell and the combination of its biophysical parameters, mainly the conductances across and along the neurites. As to why it holds, our main argument is that results based on the linear model agree with all known (to us) empirical results, and this is the simplest model.

In general, there is a clear takeaway here, which is that the dendritic tree of MBON-α3 in the lobes is highly electrotonically compact. The authors did not provide much explanation as to why this is, and the paper would benefit from a clearer conclusion. Furthermore, I found the results of Figures 4 and 5 rather straightforward given this previous observation. I am sceptical about whether the tiny variations in, e.g. Figures 3I and 5F-H, are meaningful biologically.

Please see the comment above as to the ’why’ we believe the neuron is electrotonically compact: a model with this assumption agrees well with empirically found results.

We agree that the small variations in Figure 5F-H are likely not biologically meaningful. We state this now more clearly in the figure legends and in the text. This result is important to show, however. It is precisely because these variations are small, compared to the differences between voltage differences between different numbers of activated KCs (Figure 5D) or different levels of activated synapses (Figure 5E) that we can conclude that a 25% change in either synaptic strength or number can represent clearly distinguishable internal states, and that both changes have the same effect. It is important to show these data, to allow the reader to compare the differences that DO matter (Figure 5D,E) and those that DON’T (Figure 5F-H).

The same applies to Figure 3I. The reviewer is entirely correct: the differences in the somatic voltage shown in Figure 3I are minuscule, less than a micro-Volt, and it is very unlikely that these difference have any biological meaning. The point of this figure is exactly to show this!. It is to demonstrate quantitatively the transformation of the large differences between voltages in the dendritic tree and the nearly complete uniform voltage at the soma. We feel that this shows very clearly the extreme "democratization" of the synaptic input! Please see also the reply to your second comment below.

1) My biggest question is about the claim of extreme electrotonic compactness of this neuron. Figure 3D,E suggests that the voltage change at the proximal neurite and at the soma varies by only about 1% depending on stimulation location. Since this is supported only by simulation, it is worth asking how robust this conclusion is.

The results of the simulation are very robust. The linearity of the system makes the solution of the underlying equations very stable; small perturbations in the input are guaranteed (by the nature of linear systems) to only result in small changes in the output.

As for how well the results reflect biological reality, this is a question that eventually needs to be answered by empirical investigations. The results of any theory, like the one in this study, are subject to verification/falsification. We consider our theoretical results as predictions that are easily falsifiable given suitable experimental techniques. Future empirical studies will determine the validity of these predictions.

a) Given that the variability in 3I is so small in magnitude, any dependence would be swamped by other sources of heterogeneity, so the statement that this correlates with distance (line 278) is likely irrelevant.

The reviewer is precisely right, the differences in the somatic voltage shown in Figure 3I are minuscule, less than a micro-Volt. But that is exactly the point of this Figure!

What we show in Figure 3H is that there are substantial differences (by at least an order of magnitude) between "local" voltages, i.e. at the synaptic locations in the dendritic tree. This is expected because of the much smaller diameter of distal parts of the tree than more proximal ones. What is unexpected (at least it was to us) was that these differences are EXACTLY (within a fraction of a micro Volt) compensated by the electrotonic decay from the synapses to the soma. This is the case for ALL 3121 synaptic segments in the dendritic tree. It thus appears as if the morphology of the neuron is fine-tuned to result in extreme ’democratization’ of all synapses.

Note that this goes beyond the equalizing effect resulting from a large space constant (*λ*) which is also present in this neuron (at least in our simulation; we are not aware of published results reporting empirically determined voltage distributions in the dendritic tree of this neuron). While large *λ* compresses the range of voltage distributions at distant locations, it is a linear phenomenon and as such it preserves the relative voltage values. The interplay of segment size and, more generally, morphology of the dendritic tree allows for nonlinear phenomena, such as the compression of the voltage range in the dendritic tree which vary over a factor of *≈* 30 (Figure 3H) towards a much smaller range (variation by about 1%) in Figure 3I.

Reviewer 2 had a similar concern, so we clearly did a very poor job in explaining this. We therefore rewrote the early parts of section 2. to make this more clear (line 268ff).

b) Can the authors provide a confidence interval for the fit of biophysical parameters to their recordings?

The fitting was performed as an iterative approach in the NEURON environment until we got the best fit of simulated results to neurophysiological data was achieved. Please see details in comments to reviewer 2 above. As such we cannot provide a confidence interval for the fit.

c) On lines 271-274, the authors state, "This architecture with the smallest dendritic sections at the most distant sites may contribute to the compactness of the dendritic tree, ensuring that even the most distant synaptic inputs result in somatic voltage deflections comparable to the most proximal ones." Is a dependence of dendritic size on distance required for the results? It seems like the result is simply that there is no attenuation within the dendritic tree at all. In general, the authors don't actually provide an explanation for the compactness. In the discussion, it is stated that, "The compactification of the neuron is likely related to the architectural structure of its dendritic tree," which again is rather vague. The authors should strive to provide a clear explanation for this, since it is their key result.

We agree that we should have been more explicit in describing the origin of the electrotonic compactness of the neuron. We also agree that the main reason is the low attenuation of electrical signals in the dendritic tree (large space constant λ). We now add a paragraph (second paragraph of Section 2.3) where we compute an estimate for the characteristic length of the cell and show that it substantially exceeds the cell size, resulting in low decay of distal input.

However, we believe that there is a second effect that contributes to the compactness of the cell. Analysis of the dendritic morphology reveals that the volume of dendritic segments is inversely correlated with the distance from the soma (Figure 3G). For a given current input (or synaptic conductance change), this results in much higher voltage excursions in the distal parts of the dendritic tree, with small neurite diameters, than in more proximal parts, with larger diameters, Figure 3H. As a consequence, the additional voltage decreases due to longer path length from distal synapses vs. proximal synapses is nearly exactly compensated by the increase of local dendritic voltage in the distal locations. This is shown in the nearly identical voltages experienced in the soma by distal and proximal synapses, Figure 2. Therefore, it is not only the large space constant which is important but also the interplay between small distal dendritic volumes and long distance between them and the soma.

d) Can the authors report what the electrotonic length for such a dendrite would be? How long before we expect to see a significant spread in 3E?

We now report the electronic length in the rewritten Section 2.3, see also the previous point. Our simple analytical calculation gives *λ*= 1,340*µ*m for the dendritic tree only, and *λ*= 1,510*µ*m for the whole neuron, when we base it on the mean of the diameters of all neuronal segments.

2) My other concern is that, once we assume this perfect integration, the subsequent analyses are all a straightforward consequence. In particular, Figures 4 and 5 just repeatedly convey that it doesn't matter which synapse is being activated, the effect on the somatic voltage is the same. I was particularly confused about the conclusions of 5F,G,H. The authors claim "small but significant differences" here, but practically speaking, I can't imagine any of the differences in these plots being meaningful.

It is correct that the neuron’s compactness determines its behavior. It is indeed one of our main conclusions that the influence of all synapses on the somatic voltage is very similar which, as we discuss, is important for the odor representation by KCs and its interpretation by the MBON. We document that with different analyses, e.g. those reported in the figure panels referred to by the reviewer. They could be considered redundant but we feel that they elucidate different aspects of the main message of our study. If the reviewer feels strongly that these panels are not needed, we are willing to do without them but we feel that they convey the message very clearly.

3) Reference to the data used to constrain the model is confusing.a) At various places in the manuscript references are made to in vivo recordings, but it appears that all of the recordings were done ex vivo.

We apologize for this mistake. Yes, all our recordings were performed ex vivo. We now state this explicitly throughout the manuscript. However, we also refer to some data by other groups that were collected in vivo.

b) On lines 389-392, the authors state: "Near-perfect agreement between experimentally observed and simulated voltage distributions in the dendritic tree shows that linear cable theory is an excellent model for information integration in this system." What recordings of the voltage distribution in the dendritic tree were performed?

Of course the reviewer is correct: Neither we nor, as far as we know, any other groups have recorded detailed voltage distributions in these neurons. We meant to refer to the voltage traces (not distributions) shown in Figure 2D-F that show excellent agreement between recorded and simulated data. The text has been corrected.

4) It seems like the conclusions are different than those of Gouwens and Wilson (2009), who described their reconstructed PNs as electrotonically extensive. The authors should comment on what about MBONs and PNs is different.

This is an excellent question, and it can be answered at several levels.

First, the most obvious one: why do the simulations result in neurons with different (compact vs extended) model neurons? In both cases, the model parameters used in the simulation are determined by finding the optimal fit to electrophysiological (patch clamp) data. Even the numerical methods are identical: both studies use the PRAXIS method in NEURON. In both cases, the fits obtained are in excellent agreement with the measured data. One difference is that we construct the dendritic tree from the morphological (EM) data while Gouwen and Wilson constructed synthetic dendrites since the exact morphology was not available at the time when they conducted their study. We do, however, not believe that this difference is of importance for the question at hand.

The electrotonic length *λ* of a neurite depends on three variables, its diameter a, membrane resistance r_m_ and inverse cytoplasmatic resistance r_a_: *λ*^2^ = ^ar^_2r_^m^a. For the diameter, we used the EM data in the dendritic tree and, as available, for other neurites, and the light-microscopic data where EM data was not available. Gouwen and Wilson used synthetic segments in the dendritic tree and light-microscopical data elsewhere. We could not find the exact values they used in their paper, therefore we can not say whether there are substantial differences between their values and ours. We consider it unlikely that this is the case.

There is a factor of nearly three between our best fit for cytoplasm resistance in the denominator; our value (85*Ω*m) is about three times smaller than Gouwen and Wilson’s (*≈* 250*Ω*m, depending on the chosen model neuron). There is also a factor of about five between the fits for the membrane resistance in the numerator: our value is about five times larger (*≈* 100K*Ω*cm^2^) than theirs (*≈* 20K*Ω*cm^2^). Since these numbers multiply, we find a factor of about fifteen. Assuming similar diameters, after taking the square root we find that their *λ* is about 4 times smaller than ours. Would we take their fitted values for r_m_ and r_a_, we would likewise obtain a neuron whose characteristic length is substantially smaller than its physical length, i.e.an electrotonically extended neuron. The computations are therefore consistent.

On an anatomical level, Scheffer et al., 2020 showed that neurons are commonly electrotonically compact within one anatomical compartment (as defined in their Table 1) but functionally segregated between compartments (their page 46ff). This is consistent with the electrotonically extended projection neuron studied by Gouwen and Wilson, which receives its input in one of the glomeruli and project its output in the mushroom body, clearly two different compartments. In contrast, MBON*α*-3 is entirely in the *α* lobe of the mushroom body, i.e. in one of the compartments from Table 1 in Scheffer et al., 2020 and therefore compact.

Finally, on a third level, does the difference between these neurons make sense functionally? We believe that it does. MBON*α*-3 receives input from KCs that represent odor input in a highly distributed sparse code. We have hypothesized in this study that this code is read by the MBONs by having all synapses count equally. This is not compatible with an extended structure since the impact of a synapse would depend on its location on the cell. In contrast, the PN studied by Gouwen and Wilson receives input in its glomerulus from the olfactory receptor neurons, from between 10 and 50 synapses (release sites). Indeed, their simulations showed that a volley of this size (they assumed 25 simultaneously active synapses) is required to generate a physiologically meaningful effect at the soma, with input from small numbers of synapses having little effect. An electrotonically compact neuron would faithfully transmit the effect of each individual synapse which due to stochastic variability will likely result in a high level of noise. Therefore, the electrotonically extended neuron enforces that only robust input from a large number of near-simultaneous inputs is registered at the soma.

We therefore conclude, that these two neurons have very different decoding roles which require exactly the difference in biophysical properties that are found in these two complementary studies by Wilson and Gouwen and our own. It will of course be interesting to directly evaluate this topic again in a comparative study in which both simulations depend on the EM-based reconstructions and are performed side-by-side. Only then, we can provide definitive answers – or alternatively we aim to highlight it in a review format. Therefore we kept this discussion for now to this place.